# Local Linear Convergence of Gradient Methods for Subspace Optimization via Strict Complementarity

**Ron Fisher**
Technion - Israel Institute of Technology
Haifa, Israel 3200003
`ronfisher@campus.technion.ac.il`

**Dan Garber**
Technion - Israel Institute of Technology
Haifa, Israel 3200003
`dangar@technion.ac.il`

## Abstract

We consider optimization problems in which the goal is to find a $k$-dimensional subspace of $\mathbb{R}^n$, $k << n$, which minimizes a convex and smooth loss. Such problems generalize the fundamental task of principal component analysis (PCA) to include robust and sparse counterparts, and logistic PCA for binary data, among others. This problem could be approached either via nonconvex gradient methods with highly-efficient iterations, but for which arguing about fast convergence to a global minimizer is difficult or, via a convex relaxation for which arguing about convergence to a global minimizer is straightforward, but the corresponding methods are often inefficient in high dimensions. In this work we bridge these two approaches under a strict complementarity assumption, which in particular implies that the optimal solution to the convex relaxation is unique and is also the optimal solution to the original nonconvex problem. Our main result is a proof that a natural nonconvex gradient method which is *SVD-free* and requires only a single QR-factorization of an $n \times k$ matrix per iteration, converges locally with a linear rate. We also establish linear convergence results for the nonconvex projected gradient method, and the Frank-Wolfe method when applied to the convex relaxation.

## 1 Introduction

We consider the problem of finding a $k$-dimensional subspace of $\mathbb{R}^n$, $k << n$, which minimizes a given objective function, where we identify a subspace with its corresponding projection matrix. That is, we consider the following optimization problem:

$$\min f(\mathbf{X}) \quad \text{subject to} \quad \mathbf{X} \in \mathcal{P}_{n,k} := \{\mathbf{Q}\mathbf{Q}^\top \mid \mathbf{Q} \in \mathbb{R}^{n \times k}, \ \mathbf{Q}^\top \mathbf{Q} = \mathbf{I}\}. \tag{1}$$

Throughout this work and unless stated otherwise, we assume that $f(\cdot)$ is convex, $\beta$-smooth (gradient Lipschitz) and, for ease of presentation, we also assume that the gradient $\nabla f(\cdot)$ is a symmetric matrix over the space of $n \times n$ symmetric matrices $\mathbb{S}^n$[1].

Problems of interest that fall into this model include among others robust counterparts of PCA, which are based on the smooth and convex Huber loss (see concrete examples in Section 4), logistic PCA [15], and sparse PCA [26]. Note that in Problem (1) we are interested in the low-dimensional subspace itself (as opposed to the projection of the data onto it, as in many other formulations), which is important for instance when the end goal is to perform dimension reduction, which is one of the most important applications of PCA-style methods.

---

[1]in case the gradient is not a symmetric matrix at some point $\mathbf{X} \in \mathbb{S}^n$, then denoting it by $\nabla_{\text{nonsym}} f(\mathbf{X})$, we can always take its symmetric counterpart $\nabla f(\mathbf{X}) = \frac{1}{2}(\nabla_{\text{nonsym}} f(\mathbf{X}) + \nabla_{\text{nonsym}} f(\mathbf{X})^\top)$ and, unless stated otherwise, our derivations throughout this work will remain the same

36th Conference on Neural Information Processing Systems (NeurIPS 2022).

Motivated by high-dimensional problems, we are interested in highly efficient (in particular in terms of the dimension $n$) first-order methods for Problem (1). Moreover, we are interested in establishing, at least locally, fast convergence to the global minimizer, despite the fact that Problem (1) is nonconvex. Subspace recovery/optimization problems similar to Problem (1) have received significant interest in recent years, see for instance [25, 28, 4, 17, 18, 11, 16, 22, 23] however, different from these works, our approach will not assume that $f(\cdot)$ admits a very specific structure (e.g., a linear or quadratic function), or will be based on a specific underlying statistical model. Instead, we will be interested in deterministic conditions that may hold for quite general $f(\cdot)$ (which is convex and smooth), and may render quite a wide variety of problems "well-posed" for efficient optimization.

We begin by briefly describing two natural dimension-efficient first-order methods for tackling Problem (1). One such method is the nonconvex projected gradient method which follows the dynamics:

$$\mathbf{X}_{t+1} \leftarrow \Pi_{\mathcal{P}_{n,k}}[\mathbf{X}_t - \eta_t \nabla f(\mathbf{X}_t)], \tag{2}$$

where $\Pi_{\mathcal{P}_{n,k}}[\cdot]$ denotes the Euclidean projection onto the set $\mathcal{P}_{n,k}$ (note that since this set is nonconvex, in general, the projection need not be unique), and $\eta_t > 0$ is the step-size. Given the gradient $\nabla f(\mathbf{X}_t)$, the runtime to compute $\mathbf{X}_{t+1}$ is dominated by the computation of the projection. It is well known that the Euclidean projection is given by the projection matrix which corresponds to the span of the top $k$ eigenvectors of the matrix $\mathbf{X}_t - \eta_t \nabla f(\mathbf{X}_t)$. While accurate computation of this projection requires a (thin) singular value decomposition (SVD) of a $n \times n$ matrix, which amounts to $O(n^3)$ runtime, it can also be approximated up to sufficiently small error using fast iterative methods, such as the well-known orthogonal iteration method [10] (aka subspace iteration method [24]). The orthogonal iteration method finds a $n \times k$ matrix $\mathbf{Q}$ with orthonormal columns which approximately span the subspace spanned by the $k$ leading eigenvectors of a given positive semidefinite $n \times n$ matrix $\mathbf{A}$, by repeatedly applying the iterations: $(\mathbf{Q}, \mathbf{R}) \leftarrow \text{QR-FACTORIZE}(\mathbf{AQ})$, where QR-FACTORIZE$(\cdot)$ denotes the QR factorization of a matrix, i.e., $\mathbf{Q} \in \mathbb{R}^{n \times k}$ has orthonormal columns. Every iteration of this method takes in worst case only $O(kn^2)$ time. When the gradient $\nabla f(\mathbf{X}_t)$ admits a favorable structure such as sparsity or a low-rank factorization, the runtime to approximate the projection onto $\mathcal{P}_{n,k}$ using the orthogonal iteration method could be significantly improved.

Another natural approach to tackle Problem (1) is to exploit the fact that each $\mathbf{X} \in \mathcal{P}_{n,k}$ could be factored as $\mathbf{X} = \mathbf{Q}\mathbf{Q}^\top$, $\mathbf{Q} \in \mathbb{R}^{n \times k}$ having orthonormal columns, and to apply gradient steps w.r.t. this factorization. This leads to the following dynamics, which we refer to as *Gradient Orthogonal Iteration*:

$$\mathbf{Z}_{t+1} \leftarrow \mathbf{Q}_t - \eta_t \frac{\partial f(\mathbf{Q}\mathbf{Q}^\top)}{\partial \mathbf{Q}}\Big|_{\mathbf{Q}_t} = \mathbf{Q}_t - \eta_t \nabla f(\mathbf{Q}_t \mathbf{Q}_t^\top)\mathbf{Q}_t,$$

$$(\mathbf{Q}_{t+1}, \mathbf{R}_{t+1}) \leftarrow \text{QR-FACTORIZE}(\mathbf{Z}_{t+1}), \tag{3}$$

where the QR-factorization step is required to ensure that $\mathbf{Q}_{t+1}\mathbf{Q}_{t+1}^\top$ is also a projection matrix.

As opposed to the Dynamics (2), which as discussed, an efficient implementation of will require to run a QR-based iterative method to compute the Euclidean projection onto $\mathcal{P}_{n,k}$ on each iteration, the Dynamics (3) only requires a single QR factorization per iteration, and thus, given the gradient matrix $\nabla f(\mathbf{Q}_t \mathbf{Q}_t^\top)$, the next iterate $\mathbf{Q}_{t+1}$ can be computed in overall $O(n^2 k)$ time. As mentioned above, this runtime could be further significantly improved if the multiplication $\nabla f(\mathbf{Q}_t\mathbf{Q}_t)\mathbf{Q}_t$ could be carried out faster than $O(n^2 k)$ (for instance when the gradient is sparse or admits a low-rank factorization), since all other operations require only $O(k^2 n)$ time (e.g., factorizing of $\mathbf{Z}_{t+1}$).

*Obtaining provable guarantees on the fast local convergence of the Dynamics (3) to a global optimal solution of Problem (1) is the main contribution of this work.*

While both Dynamics (2), (3) apply efficient iterations, since they are inherently nonconvex, arguing about their convergence to a global optimal solution of (1) is difficult in general. An alternative is to replace Problem (1) with a convex counterpart, for which, arguing about the convergence of first-order methods to a global optimal solution is well understood. Consider the convex set $\mathcal{F}_{n,k} = \text{conv}(\mathcal{P}_{n,k})$, where $\text{conv}(\cdot)$ denotes the convex-hull operation. $\mathcal{F}_{n,k}$ is also called the *Fantope* and it is known to admit the following important characterization: $\mathcal{F}_{n,k} = \{\mathbf{X} \in \mathbb{S}^n \mid \mathbf{I} \succeq \mathbf{X} \succeq 0, \text{Tr}(\mathbf{X}) = k\}$, where $\mathbf{A} \succeq 0$ denotes that $\mathbf{A}$ is a positive semidefinite matrix (PSD), see for instance [20]. This leads to the

convex problem:

$$\min f(\mathbf{X}) \quad \text{subject to} \quad \mathbf{X} \in \mathcal{F}_{n,k} = \{\mathbf{X} \in \mathbb{S}^n \mid \mathbf{I} \succeq \mathbf{X} \succeq 0, \text{Tr}(\mathbf{X}) = k\}. \tag{4}$$

A well known first-order method applicable to (4) is the Frank-Wolfe method (aka conditional gradient) [12], which for the convex Problem (4) follows the dynamics:

$$\mathbf{V}_t \leftarrow \underset{\mathbf{V} \in \mathcal{P}_{n,k}}{\arg\min} \text{Tr}(\mathbf{V} \nabla f(\mathbf{X}_t)), \quad \mathbf{X}_{t+1} \leftarrow (1 - \eta_t)\mathbf{X}_t + \eta_t \mathbf{V}_t, \ \eta_t \in [0,1]. \tag{5}$$

It follows from Ky Fan's maximum principle [7] that computing $\mathbf{V}_t$ amounts to computing the projection matrix onto the span of the $k$ eigenvectors corresponding to the $k$ smallest eigenvalues of $\nabla f(\mathbf{X}_t)$, and hence can be carried out efficiently using the orthogonal iterations method or similar methods, similarly to the computation of the projection in (2) discussed above. [2] Note however that the Frank-Wolfe iterates will not be, in general, low rank, and only yield a $O(1/t)$ convergence rate [12].

## 1.1 The eigengap assumption and strict complementarity

We now turn to discuss our only non-completely standard assumption on Problems (1), (4), which will underly all of our contributions, and in particular will facilitate our local linear convergence rates.

**Assumption 1** (Main assumption). *An optimal solution $\mathbf{X}^*$ to the convex Problem (4) is said to satisfy the eigen-gap assumption with parameter $\delta > 0$, if $\lambda_{n-k}(\nabla f(\mathbf{X}^*)) - \lambda_{n-k+1}(\nabla f(\mathbf{X}^*)) \geq \delta$.*

Assumption 1 in particular implies the following theorem which states that the convex relaxation (4) exactly recovers the unique and optimal solution to the nonconvex Problem (1). This is one aspect in which Assumption 1 captures "well-posed" instances of Problem (1). The proof is in the appendix.

**Theorem 1.** *If an optimal solution $\mathbf{X}^*$ to Problem (4) satisfies Assumption 1 with some parameter $\delta > 0$, then it has rank $k$, i.e., $\mathbf{X}^* \in \mathcal{P}_{n,k}$, and it is the unique optimal solution to both Problem (4) and Problem (1).*

Assumption 1 is tightly related to the convex Problem (4) through the concept of *strict-complementarity*, which is a classical concept in constrained continuous optimization theory [1]. A similar connection between an eigengap in the gradient at an optimal solution and strict complementarity has been already established in [5] for low-rank matrix optimization problems, where the underlying convex set is either the nuclear norm ball of matrices or the set of PSD matrices with unit trace. Now we establish a similar relationship for the convex relaxation (4) and the Fantope, which is slightly more involved. Let us write the Lagrangian of the convex Problem (4):

$$L(\mathbf{X}, \mathbf{Z}_1, \mathbf{Z}_2, s) = f(\mathbf{X}) - \langle \mathbf{Z}_1, \mathbf{X} \rangle - \langle \mathbf{Z}_2, \mathbf{I} - \mathbf{X} \rangle - s(\text{Tr}(\mathbf{X}) - k),$$

where the dual matrix variables $\mathbf{Z}_1, \mathbf{Z}_2$ are constrained to be PSD, i.e., $\mathbf{Z}_1 \succeq 0, \mathbf{Z}_2 \succeq 0$.

The KKT conditions state that $\mathbf{X}^*, (\mathbf{Z}_1^*, \mathbf{Z}_2^*, s^*)$ are corresponding optimal primal-dual solutions if and only if the following conditions hold:

1. $\mathbf{I} \succeq \mathbf{X}^* \succeq 0, \text{Tr}(\mathbf{X}^*) = k, \mathbf{Z}_1^* \succeq 0, \mathbf{Z}_2^* \succeq 0,$   2. $\nabla f(\mathbf{X}^*) = \mathbf{Z}_1^* - \mathbf{Z}_2^* + s^*\mathbf{I},$
3. $\langle \mathbf{Z}_1^*, \mathbf{X}^* \rangle = \langle \mathbf{Z}_2^*, \mathbf{I} - \mathbf{X}^* \rangle = 0.$

Condition 3 is known as *complementarity*. Since $\mathbf{Z}_1^*, \mathbf{Z}_2^*$ are PSD and $0 \preceq \mathbf{X}^* \preceq \mathbf{I}$, this further implies that $\mathbf{Z}_1^* \mathbf{X}^* = \mathbf{0}, \mathbf{Z}_2^*(\mathbf{I} - \mathbf{X}^*) = \mathbf{0}$, which in turn implies that

$$\text{range}(\mathbf{X}^*) \subseteq \text{nullspace}(\mathbf{Z}_1^*) \ \wedge \ \text{range}(\mathbf{I} - \mathbf{X}^*) \subseteq \text{nullspace}(\mathbf{Z}_2^*).$$

**Definition 1.** *A pair of primal-dual solutions $\mathbf{X}^*, (\mathbf{Z}_1^*, \mathbf{Z}_2^*, s^*)$ for Problem (4) is said to satisfy strict complementarity, if $\text{range}(\mathbf{X}^*) = \text{nullspace}(\mathbf{Z}_1^*) \vee \text{range}(\mathbf{I} - \mathbf{X}^*) = \text{nullspace}(\mathbf{Z}_2^*)$, which is the same as: $\text{rank}(\mathbf{Z}_1^*) = n - \text{rank}(\mathbf{X}^*) \vee \text{rank}(\mathbf{Z}_2^*) = \text{rank}(\mathbf{X}^*)$.*

**Theorem 2.** *If an optimal solution $\mathbf{X}^*$ for Problem (4) with $\text{rank}(\mathbf{X}^*) = k$ satisfies strict complementarity for some corresponding dual solution, then $\lambda_{n-k}(\nabla f(\mathbf{X}^*)) - \lambda_{n-k+1}(\nabla f(\mathbf{X}^*)) > 0$. Conversely, if an optimal solution $\mathbf{X}^*$ for Problem (4) satisfies $\lambda_{n-k}(\nabla f(\mathbf{X}^*)) - \lambda_{n-k+1}(\nabla f(\mathbf{X}^*)) > 0$, then it satisfies strict complementarity for every corresponding dual solution.*

---

[2] We note that one can also consider projection-based first-order methods for Problem (4), such as the projected gradient method, however in general, the projection onto the Fantope $\mathcal{F}_{n,k}$ will not be a low-rank matrix and hence its computation will require an expensive SVD computation (see details in the sequel).

The proof is given in the appendix. Strict complementarity has played a central role in several recent works, both for establishing linear convergence rates for first-order methods, e.g., [29, 6, 8, 5], and improving the runtime of projected gradient methods due to SVD computations, for low-rank matrix optimization problems, e.g., [9, 13].

## 1.2 Additional related work

Efficient gradient methods for low-rank nonconvex optimization have received significant interest in recent years, here we mention only a few. [2, 21] gave deterministic guarantees on the local convergence to a global minimizer of factorized gradient descent for certain low-rank optimization problems, under the mild assumption that a low-rank global minimizer exists. However, these results cannot capture constraints such as those in our Problem (1) which encode projection matrices. [4], which considers nonconvex gradient methods for low-rank statistical estimation, also considers constraints that cannot capture projection matrices as in Problem (1). An exception is a specific case they consider of linear objective functions. Moreover, even for linear functions such as the specific sparse PCA objective they consider, their analysis requires several non-trivial conditions to hold (e.g. local descent, local smoothness etc), which they only show to hold under Gaussian data.

## 1.3 Notation

Throughout this work we let $\|\cdot\|$ denote the Euclidean norm for vectors in $\mathbb{R}^n$ and the spectral norm (largest singular value) for matrices in $\mathbb{R}^{m \times n}$ or $\mathbb{S}^n$. We let $\|\cdot\|_F$ denote the Frobenius (Euclidean) norm for matrices. For a matrix $\mathbf{X} \in \mathbb{S}^n$, we let $\lambda_i(\mathbf{X})$ denote the $i$th largest eigenvalue of $\mathbf{X}$. We let $\langle \cdot, \cdot \rangle$ denote the standard inner-product for both spaces $\mathbb{R}^n$ and $\mathbb{S}^n$.

# 2 Overview of Results

## 2.1 Main result

Our main novel contribution is the proof of the following theorem regarding the local linear convergence of the gradient orthogonal iteration (3) to the optimal solution of Problems (4), (1).

**Theorem 3.** *[Local linear convergence of gradient orthogonal iteration] Suppose Assumption 1 holds true for some optimal solution $\mathbf{X}^*$ to Problem (4) with some parameter $\delta > 0$. Let $G \geq \sup_{\mathbf{X} \in \mathcal{F}_{n,k}} \|\nabla f(\mathbf{X})\|$. Consider the sequence $\{\mathbf{Q}_t\}_{t \geq 1}$ generated by Dynamics (3) with a fixed step-size $\eta_t = \eta = \frac{1}{5 \max\{\beta, G\}}$ for all $t \geq 1$, and when initialized with $\mathbf{Q}_1 \in \mathcal{P}_{n,k}$ such that $\|\mathbf{Q}_1 \mathbf{Q}_1^\top - \mathbf{X}^*\|_F \leq \min\{1, \sqrt{\frac{\delta}{2}}\} \frac{\eta \delta}{2(1 + \eta \beta)}$. Then, we have that*

$$\forall t \geq 1 : \quad f(\mathbf{Q}_t \mathbf{Q}_t^\top) - f(\mathbf{X}^*) \leq \left( f(\mathbf{Q}_1 \mathbf{Q}_1^\top) - f(\mathbf{X}^*) \right) \exp\left( -\frac{\delta(t-1)}{40 \max\{\beta, G\}} \right).$$

While, as stated above, this is not the first work to consider strict complementarity conditions for bridging convex and nonconvex methods for low-rank optimization, previous works such as [8, 5, 9, 13] consider gradient methods that rely on (nearly) exact (low-rank) SVD computations on each iteration, whereas Theorem 3 considers the more efficient *SVD-free* Dynamics (3), that requires only a single QR-factorization of an $n \times k$ matrix per iteration, which is much faster and simpler to implement. Accordingly, the proof is also considerably more challenging and requires new ideas.

## 2.2 Additional results

We also prove the following two theorems regarding the local linear convergence of the projected gradient Dynamics (2) and the Frank-Wolfe Dynamics (5). These extend the results in [9, 8] from optimization over the set of positive semidefinite matrices with unit trace to the Fantope.

**Theorem 4.** *[Local linear convergence of nonconvex PGD] Suppose Assumption 1 holds true for some optimal solution $\mathbf{X}^*$ to Problem (4) with some parameter $\delta > 0$. Consider the sequence $\{\mathbf{X}_t\}_{t \geq 1}$ generated by Dynamics (2) with a fixed step-size $\eta_t = \eta = 1/\beta$ for all $t \geq 1$, and when initialized with $\mathbf{X}_1 \in \mathcal{F}_{n,k}$ such that $\|\mathbf{X}_1 - \mathbf{X}^*\|_F \leq \frac{\delta}{4\beta}$. Then, for all $t \geq 1$ it holds that*

1. $rank(\mathbf{X}_{t+1}) = k$, and thus, given $\mathbf{X}_t$ and $\nabla f(\mathbf{X}_t)$, $\mathbf{X}_{t+1}$ can be computed via a rank-$k$ SVD,

2. $f(\mathbf{X}_t) - f(\mathbf{X}^*) \leq (f(\mathbf{X}_1) - f(\mathbf{X}^*)) \cdot \exp(-\Theta(\delta/\beta)(t-1)))$.

**Theorem 5.** *[Local linear convergence of Frank-Wolfe] Suppose Assumption 1 holds true for some optimal solution $\mathbf{X}^*$ to Problem (4) with some parameter $\delta > 0$. Consider the sequences $\{(\mathbf{X}_t, \mathbf{V}_t)\}_{t \geq 1}$ generated by Dynamics (5) when $\eta_t$ is chosen via line-search. Then, there exists $T_0 = O\left(k(\beta/\delta)^3\right)$ such that,*

$$\forall t \geq T_0 : f(\mathbf{X}_{t+1}) - f(\mathbf{X}^*) \leq \left(f(\mathbf{X}_t) - f(\mathbf{X}^*)\right)\left(1 - \min\{\frac{\delta}{12\beta}, \frac{1}{2}\}\right).$$

*Moreover, for all $t \geq 1$, the rank-$k$ matrix $\mathbf{V}_t$ satisfies $\|\mathbf{V}_t - \mathbf{X}^*\|_F^2 = O\left(\frac{\beta^2}{\delta^3}\left(f(\mathbf{X}_t) - f(\mathbf{X}^*)\right)\right)$.*

**What if Assumption 1 fails?** In case Assumption 1 does not hold or holds with negligible parameter $\delta$, not all is lost, since by considering weaker versions of Assumption 1, which consider eigen-gaps between higher eigenvalues, we can still guarantee that $\mathbf{X}^*$ (an optimal solution to Problem (4)) has low rank, and that at least the projected gradient method (when applied to Problem (4)), locally, will require only a low-rank SVD to compute the projection onto the Fantope, while guaranteeing the standard convergence rate of $O(1/t)$ (not linear rate as when Assumption 1 holds).

**Theorem 6.** *Let $\mathbf{X}^* \in \mathcal{F}_{n,k}$ be some optimal solution to Problem (4) and let $\mu_1 \geq \mu_2 \geq ...\mu_n$ denote the eigenvalues of $-\nabla f(\mathbf{X}^*)$. Let $r$ be the smallest integer such that $r \geq k$ and $\mu_r - \mu_{r+1} > 0$. Then, it holds that $rank(\mathbf{X}^*) \leq r$. Moreover, consider the projected gradient dynamics w.r.t. Problem (4) given by, $\mathbf{X}_{t+1} \leftarrow \Pi_{\mathcal{F}_{n,k}}[\mathbf{X}_t - \beta^{-1}\nabla f(\mathbf{X}_t)]$. For any $r' \in \{r, \ldots, n-1\}$, if $\|\mathbf{X}_1 - \mathbf{X}^*\|_F \leq \frac{\mu_k - \mu_{r'+1}}{4\beta}$, then it holds that,*

1. $\forall t \geq 1$, $rank(\mathbf{X}_{t+1}) \leq r'$, i.e., given $\mathbf{X}_t$ and $\nabla f(\mathbf{X}_t)$, $\mathbf{X}_{t+1}$ can be computed via a rank-$r'$ SVD.

2. $\{\mathbf{X}_t\}_{t \geq 1}$ converges with the standard PGD rate: $f(\mathbf{X}_t) - f(\mathbf{X}^*) = O(\beta\|\mathbf{X}_1 - \mathbf{X}^*\|_F^2/t)$.

**Remark 1.** *Note that via the parameter $r'$, Theorem 6 offers a flexible tradeoff between the radius of the ball in which PGD needs to be initialized in (increasing $r'$ increases the radius), and the rank of the iterates which in turn, implies an upper-bound on the rank of SVD computations required for the projection, which controls the runtime of each iteration.*

**Remark 2.** *Theorem 6 may be in particular interesting when $f(\cdot)$ is subspace-monotone in the sense that for any two subspaces $\mathcal{S}_1 \subseteq \mathcal{S}_2 \subseteq \mathbb{R}^n$ and their corresponding projection matrices $\mathbf{P}_1, \mathbf{P}_2 \in \mathbb{S}^n$, it holds that $f(\mathbf{P}_2) \leq f(\mathbf{P}_1)$. In this case, given an optimal solution $\mathbf{X}^*$ to the convex Problem (4) with eigen-decomposition $\mathbf{X}^* = \sum_{i=1}^r \lambda_i \mathbf{u}_i \mathbf{u}_i^\top$, when $k < r << n$, using a projection matrix $\mathbf{P}^* = \sum_{i=1}^r \mathbf{u}_i \mathbf{u}_i^\top$ which satisfies $f(\mathbf{P}^*) \leq \min_{\mathbf{X} \in \mathcal{P}_{n,k}} f(\mathbf{X})$ may be of interest. For instance, it is not hard to show that $f(\cdot)$ of the form $f(\mathbf{X}) = \sum_{i=1}^m g_i(\|\mathbf{q}_i - \mathbf{X}\mathbf{q}_i\|)$, where $g_i(\cdot)$ is monotone non-decreasing and $\{\mathbf{q}_i\}_{i=1}^m \subset \mathbb{R}^n$, is subspace-monotone.*

The complete proofs of Theorems 3, 4, 5, 6, as well as additional results, are given in the appendix. Below we give the main ideas in the proof of Theorem 3.

## 3  Proof Sketch of Theorem 3

### 3.1  Preliminaries

**Lemma 1** (Euclidean projection onto the Fantope). *Let $\mathbf{X} \in \mathbb{S}^n$ and consider its eigen decomposition $\mathbf{X} = \sum_{i=1}^n \gamma_i \mathbf{u}_i \mathbf{u}_i^\top$. The Euclidean projection $\Pi_{\mathcal{F}_{n,k}}[\mathbf{X}]$ is given by: $\Pi_{\mathcal{F}_{n,k}}[\mathbf{X}] = \sum_{i=1}^n \gamma_i^+(\theta)\mathbf{u}_i \mathbf{u}_i^\top$, where $\gamma_i^+(\theta) = \min(\max(\gamma_i - \theta, 0), 1)$, and $\theta$ satisfies the equation $\sum_{i=1}^n \gamma_i^+(\theta) = k$. Moreover, $\forall r \in \{k, ..., n-1\}$ it holds that $rank(\Pi_{\mathcal{F}_{n,k}}(\mathbf{X})) \leq r$ if and only if $\sum_{i=1}^r \min(\gamma_i - \gamma_{r+1}, 1) \geq k$.*

**Remark 3.** *Lemma 1 implies that if $rank(\mathbf{X}) \leq r$, then only the top $r$ components in the SVD of $\mathbf{X}$ are needed to compute $\Pi_{\mathcal{F}_{n,k}}[\mathbf{X}]$, i.e., a rank-$r$ SVD of $\mathbf{X}$. Moreover, given the rank-$(r+1)$ SVD, we can check the condition $\sum_{i=1}^r \min(\gamma_i - \gamma_{r+1}, 1) \geq k$, to verify whether the projection has rank $\leq r$.*

The following lemma lower bounds, under Assumption 1, the radius of the ball around the unique optimal solution $\mathbf{X}^*$ inside-which, the PGD mapping w.r.t. the Fantope $\mathcal{F}_{n,k}$ with a fixed step-size, is guaranteed to produce rank-$k$ matrices, i.e., matrices in $\mathcal{P}_{n,k}$, which means that it coincides precisely with the PGD mapping w.r.t. the nonconvex set $\mathcal{P}_{n,k}$, given by the Dynamics (2).

**Lemma 2.** *Let $\mathbf{X}^* \in \mathcal{F}_{n,k}$ be an optimal solution to Problem (4) which satisfies Assumption 1 with some parameter $\delta > 0$, and let $\eta > 0$. For any $\mathbf{X} \in \mathcal{F}_{n,k}$ which satisfies $\|\mathbf{X} - \mathbf{X}^*\|_F \leq \dfrac{\eta \delta}{2(1 + \eta \beta)}$, it holds that $rank(\Pi_{\mathcal{F}_{n,k}}[\mathbf{X} - \eta \nabla f(\mathbf{X})]) = k$.*

The following lemma establishes that under Assumption 1, Problem (4) has a quadratic growth property. This property is known to facilitate linear convergence rates of gradient methods [19, 14].

**Lemma 3** (Quadratic Growth). *Let $\mathbf{X}^* \in \mathcal{F}_{n,k}$ be an optimal solution to Problem (4) for which Assumption 1 holds with some $\delta > 0$. Then, $\forall \mathbf{X} \in \mathcal{F}_{n,k} : \|\mathbf{X} - \mathbf{X}^*\|_F^2 \leq \dfrac{2}{\delta}(f(\mathbf{X}) - f(\mathbf{X}^*))$.*

### 3.2 Gradient Orthogonal Iteration Analysis

We outline the proof of our main algorithmic result — the local linear convergence result of the gradient orthogonal iteration (3) given in Theorem 3. For convenience, we rewrite the Dynamics (3) as Algorithm 1 below which also introduces notation that will be helpful throughout the analysis. Throughout this section we also introduce the auxiliary sequence $\{\mathbf{X}_t\}_{t \geq 1} \subset \mathcal{F}_{n,k}$ given by: $\mathbf{X}_1 = \mathbf{Y}_1$ and $\mathbf{X}_{t+1} = \Pi_{\mathcal{F}_{n,k}}[\mathbf{Y}_t - \eta \nabla f(\mathbf{Y}_t)]$ for all $t \geq 1$.

At a high-level, our analysis of Algorithm 1 relies on the following two components:

1. Using Lemma 2 we can argue that, in the proximity of $\mathbf{X}^*$, $rank(\mathbf{X}_t) = k$, i.e., $\mathbf{X}_t \in \mathcal{P}_{n,k}$. This implies that $\mathbf{X}_t$ is the projection matrix onto the span of top $k$ eigenvectors of $\mathbf{W}_t$.

2. We view $\mathbf{Q}_t$ as the outcome of applying one iteration of the orthogonal iterations method [10, 24] to $\mathbf{W}_t$ (see also discussion in the Introduction). Combined with the previous point, this allows to argue that $\mathbf{Y}_t = \mathbf{Q}_t \mathbf{Q}_t^\top$ is sufficiently close to the projected gradient update $\mathbf{X}_t$, which drives the convergence.

---

**Algorithm 1** Gradient Orthogonal Iteration

---
1: initialization: $\mathbf{Y}_1 = \mathbf{Q}_1 \mathbf{Q}_1^\top$ for some $\mathbf{Q}_1 \in \mathbb{R}^{n \times k}$ such that $\mathbf{Q}_1^\top \mathbf{Q}_1 = \mathbf{I}$
2: **for** $t = 1, 2...$ **do**
3: $\quad$ $\mathbf{W}_{t+1} \leftarrow \mathbf{Y}_t - \eta \nabla f(\mathbf{Y}_t)$
4: $\quad$ $(\mathbf{Q}_{t+1}, \mathbf{R}_{t+1}) \leftarrow$ QR-FACTORIZE$(\mathbf{W}_{t+1}\mathbf{Q}_t)$ (that is $\mathbf{Q}_{t+1}\mathbf{R}_{t+1} = \mathbf{W}_{t+1}\mathbf{Q}_t$)
5: $\quad$ $\mathbf{Y}_{t+1} \leftarrow \mathbf{Q}_{t+1}\mathbf{Q}_{t+1}^\top$
6: **end for**

---

The following key lemma establishes the connection between the sequence $\{\mathbf{Y}_t\}_{t \geq 1}$ produced by Algorithm 1, and the corresponding sequence of exact projected gradient steps $\{\mathbf{X}_t\}_{t \geq 1}$. The proof relies on an original extension of the classical orthogonal iteration method (see [10]).

**Lemma 4.** *Fix some iteration $t \geq 1$. Suppose that $\eta < 1/G$, $\mathbf{X}_{t+1} \in \mathcal{P}_{n,k}$, and $\|\mathbf{X}_{t+1} - \mathbf{Y}_t\|_F < \sqrt{2}$. It holds that, $\|\mathbf{X}_{t+1} - \mathbf{Y}_{t+1}\|_F^2 \leq \dfrac{1}{1 - \frac{1}{2}\|\mathbf{X}_{t+1} - \mathbf{Y}_t\|_F^2} \left(\dfrac{\eta G}{1 - \eta G}\right)^2 \|\mathbf{X}_{t+1} - \mathbf{Y}_t\|_F^2$.*

*Proof of Lemma 4.* Let us write the eigen-decomposition of $\mathbf{W}_{t+1} = \mathbf{Y}_t - \eta \nabla f(\mathbf{Y}_t)$ as:

$$\mathbf{W}_{t+1} = \mathbf{V}\Lambda\mathbf{V}^\top = \begin{bmatrix} \mathbf{V}_1 & \mathbf{V}_2 \end{bmatrix} \begin{bmatrix} \Lambda_1 & 0 \\ 0 & \Lambda_2 \end{bmatrix} \begin{bmatrix} \mathbf{V}_1^\top \\ \mathbf{V}_2^\top \end{bmatrix},$$

where $\mathbf{V}_1 \in \mathbb{R}^{n \times k}, \Lambda_1 \in \mathbb{R}^{k \times k}$ correspond to the largest $k$ eigenvalues.

The main part of the proof will be to prove that $\|\mathbf{V}_2^\top \mathbf{Q}_{t+1}\|_F^2 \leq \dfrac{1}{\sigma_{\min}^2(\mathbf{V}_1^\top \mathbf{Q}_t)} \left(\dfrac{\eta G}{1 - \eta G}\right)^2 \|\mathbf{V}_2^\top \mathbf{Q}_t\|_F^2$.

Note that by definition of $\mathbf{X}_{t+1}$ we have that,

$$\mathbf{X}_{t+1} = \underset{\mathbf{X} \in \mathcal{F}_{n,k}}{\arg\min} \|\mathbf{X} - \mathbf{W}_{t+1}\|_F^2 \underset{(a)}{=} \underset{\mathbf{X} \in \mathcal{P}_{n,k}}{\arg\min} \|\mathbf{X} - \mathbf{W}_{t+1}\|_F^2 \underset{(b)}{=} \underset{\mathbf{X} \in \mathcal{P}_{n,k}}{\arg\max} \langle \mathbf{X}, \mathbf{W}_{t+1}\rangle = \mathbf{V}_1\mathbf{V}_1^\top,$$

where (a) follows from the assumption of the lemma that $\mathbf{X}_{t+1} \in \mathcal{P}_{n,k}$, and (b) follows since all matrices in $\mathcal{P}_{n,k}$ have the same Frobenius norm.

This further implies that

$$\sigma_{\min}^2(\mathbf{V}_1^\top \mathbf{Q}_t) = \lambda_k(\mathbf{V}_1^\top \mathbf{Q}_t \mathbf{Q}_t^\top \mathbf{V}_1) = \sum_{i=1}^{k} \lambda_i(\mathbf{V}_1^\top \mathbf{Q}_t \mathbf{Q}_t^\top \mathbf{V}_1) - \sum_{j=1}^{k-1} \lambda_j(\mathbf{V}_1^\top \mathbf{Q}_t \mathbf{Q}_t^\top \mathbf{V}_1)$$

$$\geq \mathrm{Tr}(\mathbf{V}_1^\top \mathbf{Q}_t \mathbf{Q}_t^\top \mathbf{V}_1) - (k-1)\lambda_1(\mathbf{V}_1^\top \mathbf{Q}_t \mathbf{Q}_t^\top \mathbf{V}_1) \geq \mathrm{Tr}(\mathbf{X}_{t+1}\mathbf{Y}_t) - (k+1)$$

$$= \left(k - \frac{1}{2}\|\mathbf{X}_{t+1} - \mathbf{Y}_t\|_F^2\right) - (k-1) = 1 - \frac{1}{2}\|\mathbf{X}_{t+1} - \mathbf{Y}_t\|_F^2. \tag{6}$$

Thus, under the assumption that $\|\mathbf{X}_{t+1} - \mathbf{Y}_t\|_F < \sqrt{2}$, we have that $(\mathbf{V}_1^\top \mathbf{Q}_t)$ is invertible.

Since $(\mathbf{Q}_{t+1}, \mathbf{R}_{t+1})$ is the QR factorization of $\mathbf{W}_{t+1}\mathbf{Q}_t$, using the eigen-decomposition of $\mathbf{W}_{t+1}$ we can write $\mathbf{Q}_{t+1}\mathbf{R}_{t+1} = \mathbf{V}\Lambda\mathbf{V}^\top \mathbf{Q}_t$. Multiplying both sides from the left by $\mathbf{V}^\top$ we get,

$$\begin{bmatrix} \mathbf{V}_1^\top \mathbf{Q}_{t+1} \\ \mathbf{V}_2^\top \mathbf{Q}_{t+1} \end{bmatrix} \mathbf{R}_{t+1} = \begin{bmatrix} \Lambda_1 & 0 \\ 0 & \Lambda_2 \end{bmatrix} \begin{bmatrix} \mathbf{V}_1^\top \mathbf{Q}_t \\ \mathbf{V}_2^\top \mathbf{Q}_t \end{bmatrix},$$

which leads to the two equations:

$$\Lambda_1 \mathbf{V}_1^\top \mathbf{Q}_t = \mathbf{V}_1^\top \mathbf{Q}_{t+1} \mathbf{R}_{t+1}, \tag{7}$$

$$\Lambda_2 \mathbf{V}_2^\top \mathbf{Q}_t = \mathbf{V}_2^\top \mathbf{Q}_{t+1} \mathbf{R}_{t+1}. \tag{8}$$

Under the assumption that $\eta < 1/G$, using Weyl's inequality we have that $\lambda_k(\mathbf{W}_{t+1}) \geq \lambda_k(\mathbf{Y}_t) - \eta\lambda_1(\nabla f(\mathbf{Y}_t)) > 0$, and so $\Lambda_1$ is invertible. Since from (6) we have that $\sigma_{\min}(\mathbf{V}_1^\top \mathbf{Q}_t) > 0$, it follows that $\mathrm{rank}(\Lambda_1 \mathbf{V}_1^\top \mathbf{Q}_t) = k$ and thus, from Equation (7) we have that $\mathbf{V}_1^\top \mathbf{Q}_{t+1}$ and $\mathbf{R}_{t+1}$ are both invertible and we can write $\mathbf{R}_{t+1} = (\mathbf{V}_1^\top \mathbf{Q}_{t+1})^{-1}\Lambda_1 \mathbf{V}_1^\top \mathbf{Q}_t$.

Multiplying both sides of Equation (8) from the right with $\mathbf{R}_{t+1}^{-1}$, we get

$$\mathbf{V}_2^\top \mathbf{Q}_{t+1} = \Lambda_2 \mathbf{V}_2^\top \mathbf{Q}_t \left((\mathbf{V}_1^\top \mathbf{Q}_{t+1})^{-1}\Lambda_1 \mathbf{V}_1^\top \mathbf{Q}_t\right)^{-1} = \Lambda_2 \mathbf{V}_2^\top \mathbf{Q}_t (\mathbf{V}_1^\top \mathbf{Q}_t)^{-1}\Lambda_1^{-1}\mathbf{V}_1^\top \mathbf{Q}_{t+1}.$$

Now we can use this to bound $\|\mathbf{V}_2^\top \mathbf{Q}_{t+1}\|_F^2$:

$$\|\mathbf{V}_2^\top \mathbf{Q}_{t+1}\|_F^2 = \|\Lambda_2 \mathbf{V}_2^\top \mathbf{Q}_t (\mathbf{V}_1^\top \mathbf{Q}_t)^{-1}\Lambda_1^{-1}\mathbf{V}_1^\top \mathbf{Q}_{t+1}\|_F^2$$

$$\underset{(a)}{\leq} \|(\mathbf{V}_1^\top \mathbf{Q}_t)^{-1}\|_2^2 \|\Lambda_1^{-1}\|_2^2 \|\mathbf{V}_1^\top \mathbf{Q}_{t+1}\|_2^2 \|\Lambda_2\|_2^2 \|\mathbf{V}_2^\top \mathbf{Q}_t\|_F^2 \underset{(b)}{\leq} \frac{\|\mathbf{V}_2^\top \mathbf{Q}_t\|_F^2}{\sigma_{\min}^2(\mathbf{V}_1^\top \mathbf{Q}_t)} \left(\frac{\lambda_{k+1}(\mathbf{W}_{t+1})}{\lambda_k(\mathbf{W}_{t+1})}\right)^2, \tag{9}$$

where (a) follows from the inequalities $\|\mathbf{A}\mathbf{B}\|_F \leq \min\{\|\mathbf{A}\|_F\|\mathbf{B}\|_2, \|\mathbf{A}\|_2\|\mathbf{B}\|_F\}$, $\|\mathbf{A}\mathbf{B}\|_2 \leq \|\mathbf{A}\|_2\|\mathbf{B}\|_2$, and (b) follows from the eigen-decomposition of $\mathbf{W}_{t+1}$ and by noting that since $\mathbf{V}_1, \mathbf{Q}_{t+1}$ both have orthonormal columns, it holds that $\|\mathbf{V}_1^\top \mathbf{Q}_{t+1}\|_2 \leq 1$.

We upper-bound $\lambda_{k+1}(\mathbf{W}_{t+1})/\lambda_k(\mathbf{W}_{t+1})$ by using Weyl's inequality as follows:

$$\frac{\lambda_{k+1}(\mathbf{W}_{t+1})}{\lambda_k(\mathbf{W}_{t+1})} \leq \frac{\lambda_{k+1}(\mathbf{Y}_t) + \eta\lambda_1(-\nabla f(\mathbf{Y}_t))}{\lambda_k(\mathbf{Y}_t) + \eta\lambda_n(-\nabla f(\mathbf{Y}_t))} \leq \frac{\eta G}{1 - \eta G}, \tag{10}$$

where we have used the fact that $\mathbf{Y}_t \in \mathcal{P}_{n,k}$, and so $\lambda_k(\mathbf{Y}_t) = 1, \lambda_{k+1}(\mathbf{Y}_t) = 0$.

Plugging (10) into (9) we indeed obtain,

$$\|\mathbf{V}_2^\top \mathbf{Q}_{t+1}\|_F^2 \leq \frac{1}{\sigma_{\min}^2(\mathbf{V}_1^\top \mathbf{Q}_t)} \left(\frac{\eta G}{1 - \eta G}\right)^2 \|\mathbf{V}_2^\top \mathbf{Q}_t\|_F^2. \tag{11}$$

Now, for the final part of the proof, we note that $\|\mathbf{V}_2^\top \mathbf{Q}_{t+1}\|_F^2 = \mathrm{Tr}(\mathbf{V}_2\mathbf{V}_2^\top \mathbf{Y}_{t+1}) = \mathrm{Tr}((\mathbf{I} - \mathbf{X}_{t+1})\mathbf{Y}_{t+1}) = k - \mathrm{Tr}(\mathbf{X}_{t+1}\mathbf{Y}_{t+1}) = \frac{1}{2}\|\mathbf{X}_{t+1} - \mathbf{Y}_{t+1}\|_F^2$, and similarly, $\|\mathbf{V}_2^\top \mathbf{Q}_t\|_F^2 = \frac{1}{2}\|\mathbf{X}_{t+1} - \mathbf{Y}_t\|_F^2$. Plugging these observations and (6) into (11), we obtain the lemma. $\square$

The following lemma is the main step in the proof of the convergence rate of Algorithm 1.

**Lemma 5.** *Let us denote $h_t = f(\mathbf{Y}_t) - f(\mathbf{X}^*)$ for all $t \geq 1$. Fix some iteration $t$ of Algorithm 1, and suppose that $\eta \leq \frac{1}{5\max\{\beta,G\}}$, $\mathbf{X}_{t+1} \in \mathcal{P}_{n,k}$, and that $\|\mathbf{X}_{t+1} - \mathbf{Y}_t\|_F \leq 1$. Denote the constants $C_0 = 2\left(\frac{\eta G}{1-\eta G}\right)^2$, $C_1 = \frac{2(1+\eta G)C_0}{1-2\eta\beta-2C_0(1+\eta G)}$. It holds that, $h_{t+1} \leq \left(1 - \frac{\eta\delta}{4(1+C_1)}\right)h_t$, where $\delta > 0$ is the constant from Assumption 1.*

*Proof.* Using the $\beta$-smoothness of $f(\mathbf{X})$, for any $\mathbf{X} \in \mathcal{F}_{n,k}$ and $\eta \leq \frac{1}{\beta}$ it holds that

$$f(\mathbf{X}) \leq f(\mathbf{Y}_t) + \langle \mathbf{X} - \mathbf{Y}_t, \nabla f(\mathbf{Y}_t)\rangle + \frac{1}{2\eta}\|\mathbf{X} - \mathbf{Y}_t\|_F^2$$

$$\underset{(a)}{\leq} f(\mathbf{Y}_t) + \langle \mathbf{X} - \mathbf{Y}_t, \nabla f(\mathbf{Y}_t)\rangle + \eta^{-1}\langle \mathbf{Y}_t, \mathbf{Y}_t - \mathbf{X}\rangle$$

$$= f(\mathbf{Y}_t) + \eta^{-1}\langle \mathbf{Y}_t - \mathbf{X}, \mathbf{Y}_t - \eta\nabla f(\mathbf{Y}_t)\rangle, \tag{12}$$

where (a) follows since using the fact that $\mathbf{Y}_t \in \mathcal{P}_{n,k}$, we have that for any $\mathbf{X} \in \mathcal{F}_{n,k}$ it holds that $\|\mathbf{X}\|_F^2 \leq k = \|\mathbf{Y}_t\|_F^2 = \langle \mathbf{Y}_t, \mathbf{Y}_t\rangle$.

Since $\mathbf{X}_{t+1} = \Pi_{\mathcal{F}_{n,k}}[\mathbf{Y}_t - \eta\nabla f(\mathbf{Y}_t)] = \arg\min_{\mathbf{X}\in\mathcal{F}_{n,k}}\|\mathbf{X} - (\mathbf{Y}_t - \eta\nabla f(\mathbf{Y}_t))\|_F^2$, and by the assumption of the lemma that $\mathbf{X}_{t+1} \in \mathcal{P}_{n,k}$, using the first-order optimality condition, it can be shown that for all $\mathbf{Z} \in \mathcal{F}_{n,k}$: $\langle \mathbf{X}_{t+1} - \mathbf{Z}, \mathbf{Y}_t - \eta_t\nabla f(\mathbf{Y}_t)\rangle \geq 0$, see Lemma 6. This implies that for all $\mathbf{Z} \in \mathcal{F}_{n,k}$:

$$\langle \mathbf{Y}_{t+1}, \mathbf{Y}_t - \eta\nabla f(\mathbf{Y}_t)\rangle = \langle \mathbf{X}_{t+1}, \mathbf{Y}_t - \eta\nabla f(\mathbf{Y}_t)\rangle - \langle \mathbf{X}_{t+1} - \mathbf{Y}_{t+1}, \mathbf{Y}_t - \eta\nabla f(\mathbf{Y}_t)\rangle \geq$$

$$\langle \mathbf{Z}, \mathbf{Y}_t - \eta\nabla f(\mathbf{Y}_t)\rangle - \langle \mathbf{X}_{t+1} - \mathbf{Y}_{t+1}, \mathbf{Y}_t - \eta\nabla f(\mathbf{Y}_t)\rangle \geq$$

$$\langle \mathbf{Z}, \mathbf{Y}_t - \eta\nabla f(\mathbf{Y}_t)\rangle - \|\mathbf{X}_{t+1} - \mathbf{Y}_{t+1}\|_F^2\|\mathbf{W}_{t+1}\|_2, \tag{13}$$

where the last inequality is due to Lemma 8, which uses again the facts that $\mathbf{X}_{t+1} \in \mathcal{P}_{n,k}$ and $\mathbf{X}_{t+1} = \arg\min_{\mathbf{X}\in\mathcal{F}_{n,k}}\|\mathbf{X} - (\mathbf{Y}_t - \eta\nabla f(\mathbf{Y}_t))\|_F^2$, which in turn imply that $\mathbf{X}_{t+1} = \arg\max_{\mathbf{X}\in\mathcal{P}_{n,k}}\langle\mathbf{X}, \mathbf{W}_{t+1}\rangle$, and recalling that $\mathbf{W}_{t+1} = \mathbf{Y}_t - \eta\nabla f(\mathbf{Y}_t)$.

Setting $\mathbf{X} = \mathbf{Y}_{t+1}$ in (12) and plugging-in (13), we have that for any $\mathbf{Z} \in \mathcal{F}_{n,k}$ it holds that,

$$f(\mathbf{Y}_{t+1}) \leq f(\mathbf{Y}_t) + \eta^{-1}\left(\langle \mathbf{Y}_t - \mathbf{Z}, \mathbf{Y}_t - \eta\nabla f(\mathbf{Y}_t)\rangle + \|\mathbf{X}_{t+1} - \mathbf{Y}_{t+1}\|_F^2\|\mathbf{W}_{t+1}\|_2\right)$$

$$= f(\mathbf{Y}_t) + \langle \mathbf{Z} - \mathbf{Y}_t, \nabla f(\mathbf{Y}_t)\rangle + \frac{1}{2\eta}\|\mathbf{Z} - \mathbf{Y}_t\|_F^2 + \frac{1}{\eta}\|\mathbf{X}_{t+1} - \mathbf{Y}_{t+1}\|_F^2\|\mathbf{W}_{t+1}\|_2$$

$$\leq f(\mathbf{Y}_t) + \langle \mathbf{Z} - \mathbf{Y}_t, \nabla f(\mathbf{Y}_t)\rangle + \frac{1}{2\eta}\|\mathbf{Z} - \mathbf{Y}_t\|_F^2 + \frac{1+\eta G}{\eta}\|\mathbf{X}_{t+1} - \mathbf{Y}_{t+1}\|_F^2, \quad (14)$$

where the last inequality is due to the following upper-bound on $\|\mathbf{W}_{t+1}\|_2$:

$$\|\mathbf{W}_{t+1}\|_2 = \|\mathbf{Y}_t - \eta\nabla f(\mathbf{Y}_t)\|_2 \leq \|\mathbf{Y}_t\|_2 + \eta\|\nabla f(\mathbf{Y}_t)\|_2 \leq 1 + \eta G.$$

In particular, setting $\mathbf{Z} = (1-\alpha)\mathbf{Y}_t + \alpha\mathbf{X}^*$ for some $\alpha \in [0,1]$, we get that

$$f(\mathbf{Y}_{t+1}) \leq f(\mathbf{Y}_t) + \alpha\langle \mathbf{X}^* - \mathbf{Y}_t, \nabla f(\mathbf{Y}_t)\rangle + \frac{\alpha^2}{2\eta}\|\mathbf{X}^* - \mathbf{Y}_t\|_F^2 + \frac{1+\eta G}{\eta}\|\mathbf{X}_{t+1} - \mathbf{Y}_{t+1}\|_F^2.$$

Subtracting $f(\mathbf{X}^*)$ from both sides, using the convexity of $f(\cdot)$, and Lemma 3 gives

$$h_{t+1} \leq \left(1 - \alpha + \frac{\alpha^2}{\eta\delta}\right)h_t + \frac{1+\eta G}{\eta}\|\mathbf{X}_{t+1} - \mathbf{Y}_{t+1}\|_F^2.$$

Setting $\alpha = \eta\delta/2$ (note that since $\eta \leq 1/G$, we have that $\alpha \in [0,1]$), gives

$$h_{t+1} \leq \left(1 - \frac{\eta\delta}{4}\right)h_t + \frac{1+\eta G}{\eta}\|\mathbf{X}_{t+1} - \mathbf{Y}_{t+1}\|_F^2. \tag{15}$$

We now continue to upper-bound the term $\|\mathbf{X}_{t+1} - \mathbf{Y}_{t+1}\|_F^2$. Using Lemma 9, which apply standard arguments in the analysis of first-order methods, that rely only on the facts that $\mathbf{X}_{t+1} = \Pi_{\mathcal{F}_{n,k}}[\mathbf{Y}_t - \eta\nabla f(\mathbf{Y}_t)]$ and that $f(\cdot)$ is smooth and convex, we have that

$$\|\mathbf{X}_{t+1} - \mathbf{Y}_t\|_F^2 \leq \frac{\eta}{1-\eta\beta}\left(f(\mathbf{Y}_t) - f(\mathbf{X}_{t+1})\right). \tag{16}$$

Let us set $\mathbf{Z} = \mathbf{X}_{t+1}$ in (14) to obtain that

$$f(\mathbf{Y}_{t+1}) \leq f(\mathbf{Y}_t) + \langle \mathbf{X}_{t+1} - \mathbf{Y}_t, \nabla f(\mathbf{Y}_t) \rangle + \frac{1}{2\eta} \|\mathbf{X}_{t+1} - \mathbf{Y}_t\|_F^2 + \frac{1+\eta G}{\eta} \|\mathbf{X}_{t+1} - \mathbf{Y}_{t+1}\|_F^2$$

$$\leq f(\mathbf{X}_{t+1}) + \frac{1}{2\eta} \|\mathbf{X}_{t+1} - \mathbf{Y}_t\|_F^2 + \frac{1+\eta G}{\eta} \|\mathbf{X}_{t+1} - \mathbf{Y}_{t+1}\|_F^2,$$

where the last inequality is due to convexity of $f(\cdot)$. Rearranging and using Lemma 4 along with the notation $C_0 = 2\left(\frac{\eta G}{1-\eta G}\right)^2$, we have $f(\mathbf{X}_{t+1}) \geq f(\mathbf{Y}_{t+1}) - \frac{1}{\eta}\left(\frac{1}{2} + C_0(1+\eta G)\right)\|\mathbf{X}_{t+1} - \mathbf{Y}_t\|_F^2$. Plugging into (16) we obtain

$$\|\mathbf{X}_{t+1} - \mathbf{Y}_t\|_F^2 \leq \frac{\eta}{1-\eta\beta}\left(f(\mathbf{Y}_t) - f(\mathbf{Y}_{t+1}) + \frac{1}{\eta}\left(\frac{1}{2} + C_0(1+\eta G)\right)\|\mathbf{X}_{t+1} - \mathbf{Y}_t\|_F^2\right),$$

and rearranging we obtain

$$\|\mathbf{X}_{t+1} - \mathbf{Y}_t\|_F^2 \leq \frac{1}{1 - \frac{1+2C_0(1+\eta G)}{2(1-\eta\beta)}}\frac{\eta}{1-\eta\beta}\left(f(\mathbf{Y}_t) - f(\mathbf{Y}_{t+1})\right) = \frac{2\eta(h_t - h_{t+1})}{2(1-\eta\beta) - 1 - 2C_0(1+\eta G)}.$$

Using Lemma 4 again we have, $\|\mathbf{X}_{t+1} - \mathbf{Y}_{t+1}\|_F^2 \leq \frac{2\eta C_0}{1-2\eta\beta - 2C_0(1+\eta G)}(h_t - h_{t+1})$. Plugging back into (15) we obtain $h_{t+1} \leq \left(1 - \frac{\eta\delta}{4}\right)h_t + \frac{2(1+\eta G)C_0}{1-2\eta\beta-2C_0(1+\eta G)}(h_t - h_{t+1})$. Denoting $C_1 = \frac{2(1+\eta G)C_0}{1-2\eta\beta-2C_0(1+\eta G)}$, we finally obtain $h_{t+1} \leq \frac{1}{1+C_1}\left(1 - \frac{\eta\delta}{4} + C_1\right)h_t = \left(1 - \frac{\eta\delta}{4(1+C_1)}\right)h_t$, as required. The only thing left is to choose a feasible step size. We have to require: $1 - 2\eta\beta - 2C_0(1 + \eta G) > 0$. The latter holds for any $\eta \leq \frac{1}{5\max\{\beta, G\}}$. $\qquad\square$

## 4 Numerical Simulations

Due to lack of space, some of the implementation details and results are deferred to the appendix. We consider two models for robust recovery of a low-dimensional subspace from noisy samples: 1. a *spiked covariance* model, and 2. a *sparsely corrupted entries* model. In both models we minimize a robust loss based on the Huber function, which is convex and smooth, over the Fantope. We generate random instances and solve them to high precision (duality gap $< 10^{-10}$) and take the resulting point $\mathbf{X}^*$ as the optimal solution. We measure the eigen-gap in $\nabla f(\mathbf{X}^*)$ (as in Assumption 1), and we compare the recovery error w.r.t. the naive PCA solution $\mathbf{X}_{PCA}$ which simply computes the leading subspace of the empirical covariance. The results are given in Table 1. For both models the recovery error is significantly lower than that of naive PCA, which demonstrates the usefulness of the chosen models . We see that the eigen-gap assumption indeed holds with substantial values of $\delta$.

| Noise prob. $(p)$ | 0.05 | 0.1 | 0.2 | 0.3 | 0.4 | 0.5 |
|---|---|---|---|---|---|---|
| ↓ Model 1: spiked covariance ↓ | | | | | | |
| Eigen-gap $(\delta)$ | 3.21 | 2.87 | 2.36 | 2.04 | 1.501 | 1.03 |
| $\|\mathbf{X}^* - \mathbf{P}\|_F$ | 0.0047 | 0.0075 | 0.012 | 0.016 | 0.022 | 0.0298 |
| $\|\mathbf{X}_{PCA} - \mathbf{P}\|_F$ | 0.045 | 0.072 | 0.115 | 0.157 | 0.212 | 0.292 |
| ↓ Model 2: sparsely corrupted entries ↓ | | | | | | |
| Eigen-gap $(\delta)$ | 5.72 | 5.49 | 5.15 | 4.81 | 4.38 | 3.79 |
| $\|\mathbf{X}^* - \mathbf{P}\|_F$ | 0.049 | 0.067 | 0.097 | 0.111 | 0.134 | 0.148 |
| $\|\mathbf{X}_{PCA} - \mathbf{P}\|_F$ | 0.148 | 0.199 | 0.291 | 0.335 | 0.401 | 0.439 |

Table 1: Recovery and eigen-gap results for the spiked covariance and sparsely corrupted entries models with varying noise probabilities. $\mathbf{P}$ is the projection matrix onto the ground truth subspace. $n = 100$, $k = 10$, sample size $m = 500$. Results are averages of 20 i.i.d. experiments.

We additionally test the empirical convergence of nonconvex PGD (Dynamics (2)) and the gradient orthogonal iteration method (GOI, Dynamics (3)) on the two models. We initialize both methods with the PCA solution $\mathbf{X}_{PCA}$ and use the same fixed step-size for both. We examine the convergence

of both methods in terms of recovery error and approximation error (w.r.t. the objective function). Additionally, to showcase the benefit of avoiding exact SVD computations (as employed by nonconvex PGD) and using only a single QR factorization per iteration (as in GOI), we compare the runtimes of GOI and nonconvex PGD, but we exclude the time it takes to compute the gradient on each iteration and only account for the time it takes to perform either a rank $-k$ SVD or a QR factorization, where both algorithms were implemented in Python and we have used the built-in functions NUMPY.LINALG.EIGH and NUMPY.LINALG.QR to compute thin-SVDs and QR factorizations, respectively. Finally, we verify during the run of nonconvex PGD, that on each iteration, the projection onto $\mathcal{P}_{n,k}$ is indeed the same as the projection onto the Fantope $\mathcal{F}_{n,k}$ (see Remark 3), which suggests that nonconvex PGD indeed converges to the global minimum.

The results for the spiked covariance model are given in Figure 1 (the results for the sparsely corrupted entries model are very similar and given in the appendix). It can be seen that indeed the distance between the iterates of the two methods decays very quickly and so the graphs of the recovery and approximation errors of both methods coincide. We see that both methods demonstrate a linear convergence rate (w.r.t. the objective). We also see the significant savings in runtime when replacing a thin-SVD computation (in nonconvex PGD) with only a single QR factorization (in GOI).

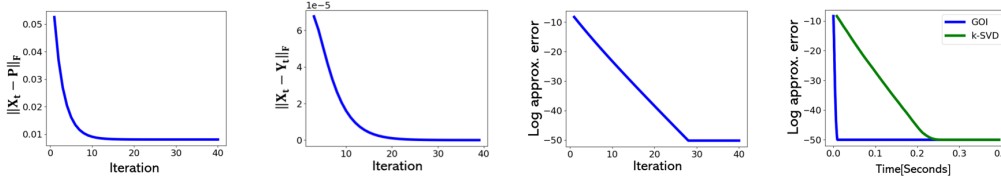

Figure 1: Convergence of PGD and GOI for the spiked covariance model with $p = 0.1$. 1st and 3rd panels from the left show the recovery error ($\mathbf{P}$ is the ground truth projection matrix) and approximation error w.r.t. objective value of PGD, respectively. Convergence of GOI is omitted since it coincides with that of PGD. 2nd panel from the left shows the distance (in Frobenius norm) between the iterates of PGD ($\mathbf{X}_t$) and those of GOI ($\mathbf{Y}_t$). The rightmost panel shows the approximation error (in log scale) vs. time, when only the time to compute matrix factorizations is taken into account.

**Importance of warm-start initalization:** We examine the performance of nonconvex PGD over $\mathcal{P}_{n,k}$ (Dynamics (2)) for the spiked covariance model considered above, but this time, when initialized with a random (uniformly distributed) projection matrix. We compare it with convex PGD which optimizes over the Fantope $\mathcal{F}_{n,k}$ and uses a full-rank SVD to compute the projection. We use the same step-size as before. We see in Figure 2 (right panel) two trends. First, we clearly see that PGD w.r.t. $\mathcal{P}_{n,k}$ and $\mathcal{F}_{n,k}$ produce very different iterates which in particular implies that, as opposed to the case of warm-start initializaion, the projections of convex PGD onto the Fantope, throughout most of the run are not rank-$k$. Second, we see that nonconvex PGD is significantly slower than convex PGD. Thus, while both methods eventually converge to the same error, this suggests that far from a global minimizer, the behaviour of nonconvex gradient methods is indeed significantly different than in the local proximity of one, which supports the fact that our theoretical guarantees only hold in a local neighbourhood of a minimizer.

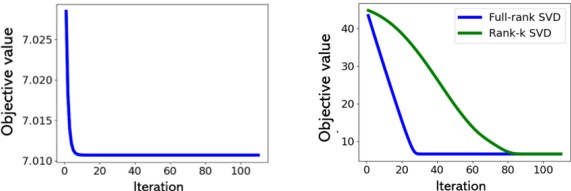

Figure 2: Convergence of PGD for the spiked covariance model with $p = 0.1$ over the Fantope $\mathcal{F}_{n,k}$ with a full-rank SVD, and over $\mathcal{P}_{n,k}$ with rank-$k$ SVD, when initialized with the PCA solution (left panel) and with random initialization (right panel). In the left panel the plots exactly coincide.

## Acknowledgements

This research was supported by the ISRAEL SCIENCE FOUNDATION (grant No. 2267/22).

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
