## A    Additional Details on Experiments

The first robust recovery model we consider is a spiked covariance model, in which we draw a uniformly distributed projection matrix onto a $k$-dimensional subspace $\mathbf{P} \in \mathcal{P}_{n,k}$, and we generate $m$ samples $\mathbf{q}_1, \ldots, \mathbf{q}_m \in \mathbb{R}^n$ such that for each $i \in [m]$, we set $\mathbf{q}_i = \mathbf{P}\mathbf{z}_i / \|\mathbf{P}\mathbf{z}_i\|$ with probability $1 - p$, and $\mathbf{q}_i = \mathbf{z}_i$ with probability $p$, where $p \in (0, 0.5)$, and $\mathbf{z}_1, \ldots, \mathbf{z}_m$ are i.i.d. uniformly distributed unit vectors. The goal is to recover $\mathbf{P}$ by minimizing the following objective function over $\mathcal{F}_{n,k}$:

$$f(\mathbf{X}) = \sum_{i=1}^{m} \mathrm{Huber}_\gamma(\|\mathbf{q}_i - a\mathbf{X}\mathbf{q}_i\|), \quad \mathrm{Huber}_\gamma(x) := \begin{cases} \frac{1}{2}x^2 & \text{if } |x| \leq \gamma \\ \gamma(|x| - \frac{1}{2}\gamma) & \text{else.} \end{cases}$$

Here $a \in (0, 1]$ is a regularization parameter and we set it to slightly less than one.

The second model we consider is that of sparsely corrupted entries in which we again draw a uniformly distributed projection matrix $\mathbf{P}$. This time the data points $\mathbf{q}_1, \ldots, \mathbf{q}_m$ are generated by taking $\mathbf{q}_i = \mathbf{P}\mathbf{z}_i / \|\mathbf{P}\mathbf{z}_i\|$ for each $i \in [m]$, where as before $\mathbf{z}_1, \ldots, \mathbf{z}_m$ are i.i.d. uniformly distributed unit vectors, but for each $i \in [m]$, with probability $p$, we pick a uniformly distributed entry $j \in [n]$ and set it to $-1$ or $+1$ (with equal probability). The goal is to recover $\mathbf{P}$ by minimizing the following objective function over $\mathcal{F}_{n,k}$:

$$f(\mathbf{X}) = \sum_{i=1}^{m} \sum_{j=1}^{n} \mathrm{Huber}_\gamma([\mathbf{q}_i]_j - [a\mathbf{X}\mathbf{q}_i]_j),$$

where here also $a \in (0, 1]$ is a regularization parameter.

For both models we set the Huber loss parameter to $\gamma = 0.1$. For the first model we set $a = 0.9$ and for the second $a = 0.8$. For a given projection matrix $\mathbf{X} \in \mathcal{P}_{n,k}$, we measure the recovery error according to $\|\mathbf{X} - \mathbf{P}\|_F^2$. For both models we let $\mathbf{X}_{PCA} \in \mathcal{P}_{n,k}$ denote the standard PCA solution, i.e., the projection matrix onto the span of the top $k$ eigenvectors of the empirical covariance $\frac{1}{m}\sum_{i=1}^{m} \mathbf{q}_i\mathbf{q}_i^\top$. For both models we set $n = 100$, $k = 10$, and $m = 500$. For both models we use the projected gradient method to find a projection matrix $\mathbf{X}^* \in \mathcal{P}_{n,k}$ which has negligible dual gap ($< 10^{-10}$). [3] For this $\mathbf{X}^*$ we measure the corresponding eigen-gap $\lambda_{n-k}(\nabla f(\mathbf{X}^*)) - \lambda_{n-k+1}(\nabla f(\mathbf{X}^*))$, and the recovery error $\|\mathbf{X}^* - \mathbf{P}\|_F$. The results are given in Table 1. For each set of parameters the results are the average of 20 i.i.d. experiments.

In a second experiment we fix for both models $p = 0.1$ and vary the dimension $n$ (while keeping $k, m$ fixed as before). The results are given in Table 2. In particular, we see that the eigen-gap $\delta$ does not change substantially with the dimension.

We turn to demonstrate the empirical performance of the projected gradient method w.r.t. to the nonconvex set $\mathcal{P}_{n,k}$ (PGD), as given in Dynamics (2), and and gradient orthogonal iteration (GOI), as given in Dynamics (3), for the two models discussed above. We fix $n = 100$ and $p = 0.1$ (keeping all other parameters unchanged). For both methods we use the fixed step-size $\eta = 1/\lambda$, where

---

[3]For $\mathbf{X} \in \mathcal{F}_{n,k}$ the dual gap is defined as $\mathrm{dg}(\mathbf{X}) = \langle \mathbf{X} - \mathbf{V}, \nabla f(\mathbf{X}) \rangle$, were $\mathbf{V} \in \arg\min_{\mathbf{Z} \in \mathcal{P}_{n,k}} \langle \mathbf{Z}, \nabla f(\mathbf{X}) \rangle$. Since $f(\cdot)$ is convex, we in particular have $f(\mathbf{X}) - \min_{\mathbf{Y} \in \mathcal{F}_{n,k}} f(\mathbf{Y}) \leq \mathrm{dg}(\mathbf{X})$.

| dim. ($n$) | 100 | 200 | 300 | 400 |
|---|---|---|---|---|
| ↓ Model 1: spiked covariance ↓ | | | | |
| Eigen-gap ($\delta$) | 2.87 | 3.02 | 2.96 | 3.04 |
| $\|\mathbf{X}^* - \mathbf{P}\|_F$ | 0.0071 | 0.005 | 0.0043 | 0.0035 |
| $\|\mathbf{X}_{PCA} - \mathbf{P}\|_F$ | 0.068 | 0.049 | 0.043 | 0.036 |
| ↓ Model 2: sparsely corrupted entries ↓ | | | | |
| Eigen-gap ($\delta$) | 5.49 | 5.902 | 6.06 | 6.1 |
| $\|\mathbf{X}^* - \mathbf{P}\|_F$ | 0.067 | 0.0617 | 0.058 | 0.055 |
| $\|\mathbf{X}_{PCA} - \mathbf{P}\|_F$ | 0.199 | 0.208 | 0.206 | 0.202 |

Table 2: Recovery and eigen-gap results for the spiked covariance model and corrupted entries model with varying dimension. Each result is the average of 20 i.i.d. experiments.

$\lambda = \lambda_1(\sum_{i=1}^m \mathbf{q}_i\mathbf{q}_i^\top)$, i.e., the largest eigenvalue of the (unnormalized) empirical covariance. We note that smaller values of $\eta$ seem too conservative in practice from our experimentations. We initialize both methods with the $k$-PCA projection matrix $\mathbf{X}_{PCA}$. We examine the convergence of both methods in terms of recovery error and approximation error (w.r.t. the objective function). Additionally, to showcase the benefit of avoiding exact SVD computations (as employed by nonconvex PGD) and using only a single QR factorization per iteration (as in GOI), we compare the runtimes of GOI and nonconvex PGD, but we exclude the time it takes to compute the gradient on each iteration and only account for the time it takes to perform either a rank $- k$ SVD or a QR factorization, where both algorithms were implemented in Python and we have used the built-in functions NUMPY.LINALG.EIGH and NUMPY.LINALG.QR to compute thin-SVDs and QR factorizations, respectively.

The results for the spiked covariance model are given in Figure 1, and the results for the sparsely corrupted entries model, which are very similar, are given in Figure 3. It can be seen that indeed the distance between the iterates of the two methods decays very quickly and so the graphs of the recovery and approximation errors of both methods coincide. We in particular see that both methods indeed demonstrate a linear convergence rate (w.r.t. the objective value). We also see the significant savings in runtime when replacing a thin-SVD computation (as used by nonconvex PGD) with only a single QR factorization (as used by GOI).

Moreover, in order to verify the convergence of nonconvex PGD to the global optimal solution (and not just a stationary point of the nonconvex Problem (1)), we verify using the procedure suggested in Remark 3, that on each iteration $t$, the projection step onto the Fantope $\mathcal{F}_{n,k}$ is also of rank $k$, i.e., identical to the projection onto $\mathcal{P}_{n,k}$. This means that the iterates of PGD w.r.t. the nonconvex Problem (1) and the iterates of PGD w.r.t. the convex relaxation (4), coincide. Indeed, for all random instances generated and for all iterations executed, we observe that the projection onto the Fantope is of rank $k$. This suggests that the nonconvex PGD (and consequently also GOI) in particular converges to the global optimal solution of the convex relaxation (4).

## B  Proof of Theorem 2

*Proof.* First, observe that for any dual solution $(\mathbf{Z}_1^*, \mathbf{Z}_2^*, s^*)$, it holds that $\mathbf{Z}_1^*$ and $\mathbf{Z}_2^*$ are orthogonal to each other. This is true since, denoting by $\mathbf{X}^*$ the corresponding primal solution, we have that,

$$\langle \mathbf{Z}_1^*, \mathbf{Z}_2^* \rangle = \mathrm{Tr}(\mathbf{Z}_1^*\mathbf{Z}_2^*) = \mathrm{Tr}(\mathbf{Z}_1^*(\mathbf{X}^* + (\mathbf{I} - \mathbf{X}^*))\mathbf{Z}_2^*)$$
$$= \mathrm{Tr}(\mathbf{Z}_1^*\mathbf{X}^*\mathbf{Z}_2^*) + \mathrm{Tr}(\mathbf{Z}_1^*(\mathbf{I} - \mathbf{X}^*)\mathbf{Z}_2^*) = 0,$$

where the last equality follows from the complementarity conditions $\mathbf{Z}_1^*\mathbf{X}^* = \mathbf{0}$ and $(\mathbf{I} - \mathbf{X}^*)\mathbf{Z}_2^* = \mathbf{0}$.

For a given dual solution $(\mathbf{Z}_1^*, \mathbf{Z}_2^*, s^*)$, let us denote $r_1 = \mathrm{rank}(\mathbf{Z}_1^*)$ and $r_2 = \mathrm{rank}(\mathbf{Z}_2^*)$.

Let us write the eigen decompositions $\mathbf{Z}_1^* = \sum_{i=1}^{r_1} \rho_i \mathbf{u}_i\mathbf{u}_i^T$ and $\mathbf{Z}_2^* = \sum_{j=1}^{r_2} \mu_j \mathbf{v}_j\mathbf{v}_j^T$.

From the orthogonality of $\mathbf{Z}_1^*$ and $\mathbf{Z}_2^*$ established above, we get an orthonormal set of vectors $\{\mathbf{u}_1, ..., \mathbf{u}_{r_1}, \mathbf{v}_1, ..., \mathbf{v}_{r_2}\}$ and we can complete it to an orthonormal basis of $\mathbb{R}^n$:

$$B = \{\mathbf{u}_1, ..., \mathbf{u}_{r_1}, \mathbf{v}_1, ..., \mathbf{v}_{r_2}, \mathbf{w}_1, ..., \mathbf{w}_{n-r_1-r_2}\},$$

where $\mathbf{Z}_1^*\mathbf{w}_i = \mathbf{0}$ and $\mathbf{Z}_2^*\mathbf{w}_i = \mathbf{0}$ for any $i \in \{1, \ldots, n - r_1 - r_2\}$.

From the KKT conditions for Problem (4), we have that $\nabla f(\mathbf{X}^*) = \mathbf{Z}_1^* - \mathbf{Z}_2^* + s^*\mathbf{I}$, and so it follows that any $\mathbf{v} \in B$ is an eigenvector of $\nabla f(\mathbf{X}^*)$. Thus, we can write the eigenvalues of $\nabla f(\mathbf{X}^*)$ in non-increasing order from left to right as:

$$\rho_1 + s^*, ..., \rho_{r_1} + s^*, \quad \underbrace{s^*, ..., s^*}_{n - r_1 - r_2 \text{ times}}, s^* - \mu_{r_2}, ..., s^* - \mu_1. \tag{17}$$

For the first direction of the theorem, let us assume $\mathbf{X}^*$ satisfies strict complementarity, so for some dual solution $(\mathbf{Z}_1^*, \mathbf{Z}_2^*, s^*)$ we have that $r_1 = n - k$ or $r_2 = k$.

Now, if $r_1 = n - k$, using (17) we get that $\lambda_{n-k}(\nabla f(\mathbf{X}^*)) = \rho_{n-k} + s^*$ and $\lambda_{n-k+1}(\nabla f(\mathbf{X}^*)) \leq s^*$, and so there is a gap of

$$\lambda_{n-k}(\nabla f(\mathbf{X}^*)) - \lambda_{n-k+1}(\nabla f(\mathbf{X}^*)) \geq \rho_{n-k} + s^* - s^* = \rho_{n-k} > 0.$$

Otherwise, if $r_2 = k$, then using (17) we have that $\lambda_{n-k+1}(\nabla f(\mathbf{X}^*)) = s^* - \mu_k$ and $\lambda_{n-k}(\nabla f(\mathbf{X}^*)) > s^*$, and so there is a gap of

$$\lambda_{n-k}(\nabla f(\mathbf{X}^*)) - \lambda_{n-k+1}(\nabla f(\mathbf{X}^*)) \geq s^* - (s^* - \mu_k) = \mu_k > 0.$$

In both cases we get a positive eigen-gap, which proves the first direction of the theorem.

For the reversed direction, let us assume that $\mathbf{X}^*$ satisfies the eigen-gap assumption, and recall that according to Theorem 1 it follows that $\text{rank}(\mathbf{X}^*) = k$. Suppose by way of contradiction that there exists a dual solution $(\mathbf{Z}_1^*, \mathbf{Z}_2^*, s^*)$ for which $r_1 < n - k$ and $r_2 < k$. In this case we have from (17) that,

$$\lambda_{n-k}(\nabla f(\mathbf{X}^*)) = \lambda_{n-k+1}(\nabla f(\mathbf{X}^*)) = s^*,$$

which contradicts the existence of an eigen-gap and so, it must be that $r_1 = n - k$ or $r_2 = k$. $\qquad\square$

## C  Details Missing from Section 3.1 and Proof of Theorem 1

### C.1  Proof of Lemma 1

*Proof.* The first part of the lemma is a known fact, see for instance [26]. For the second part, let us prove that if $\sum_{i=1}^r \min(\gamma_i - \gamma_{r+1}, 1) \geq k$, then $\theta$ must satisfy $\theta \geq \gamma_{r+1}$. Assume by way of contradiction that $\theta < \gamma_{r+1}$. Then,

$$k = \sum_{i=1}^n \min(\max(\gamma_i - \theta, 0), 1) > \sum_{i=1}^n \min(\max(\gamma_i - \gamma_{r+1}, 0), 1) = \sum_{i=1}^r \min(\gamma_i - \gamma_{r+1}, 1),$$

which is a contradiction, and so it must be that $\theta \geq \gamma_{r+1}$, and in that case the projection sets all the bottom $n - r$ components of the eigen-decomposition of $\mathbf{X}$ to zero. Hence, $\text{rank}(\Pi_{\mathcal{F}_{n,k}}[\mathbf{X}]) \leq r$. The reversed direction holds from similar reasoning. $\qquad\square$

### C.2  Proof of Theorem 1

Before we prove Theorem 1 we need the following lemma which is central to our analysis and connects between an optimal solution and the eigen-decomposition of its corresponding gradient.

**Lemma 6.** *Let $\mathbf{X}^* \in \mathcal{F}_{n,k}$ be an optimal solution to Problem (4) and write the eigen-decomposition of $-\nabla f(\mathbf{X}^*)$ as $-\nabla f(\mathbf{X}^*) = \sum_{i=1}^n \mu_i \mathbf{u}_i \mathbf{u}_i^\top$. Let $r$ be the smallest integer such that $r \geq k$ and $\mu_k - \mu_{r+1} > 0$. Then, for all $n \geq i \geq r + 1$, $\mathbf{X}^*$ is orthogonal to $\mathbf{u}_i \mathbf{u}_i^\top$, and $\text{rank}(\mathbf{X}^*) \leq r$.*
*In particular, if $r = k$, then $\mathbf{X}^* \in \mathcal{P}_{n,k}$ is the unique projection matrix onto the span of the $k$ leading eigenvectors of $-\nabla f(\mathbf{X}^*)$.*

*Proof.* Assume by way of contradiction that $\mathbf{X}^*$ is not orthogonal $\mathbf{u}_{r+1}\mathbf{u}_{r+1}^\top, \ldots, \mathbf{u}_n\mathbf{u}_n^\top$. In this case, $\sum_{i=r+1}^n \mathbf{u}_i^\top \mathbf{X}^* \mathbf{u}_i > 0$, and we can write,

$$
\begin{aligned}
\langle \mathbf{X}^*, -\nabla f(\mathbf{X}^*) \rangle &= \sum_{i=1}^r \mu_i \mathbf{u}_i^\top \mathbf{X}^* \mathbf{u}_i + \sum_{i=r+1}^n \mu_i \mathbf{u}_i^\top \mathbf{X}^* \mathbf{u}_i \\
&\underset{(a)}{<} \sum_{i=1}^r \mu_i \mathbf{u}_i^\top \mathbf{X}^* \mathbf{u}_i + \mu_r \sum_{i=r+1}^n \mathbf{u}_i^\top \mathbf{X}^* \mathbf{u}_i \\
&\underset{(b)}{=} \sum_{i=1}^{k-1} \mu_i \mathbf{u}_i^\top \mathbf{X}^* \mathbf{u}_i + \mu_k \sum_{i=k}^n \mathbf{u}_i^\top \mathbf{X}^* \mathbf{u}_i \\
&\underset{(c)}{=} \sum_{i=1}^{k-1} \mu_i \mathbf{u}_i^\top \mathbf{X}^* \mathbf{u}_i + \mu_k \left( k - \sum_{i=1}^{k-1} \mathbf{u}_i^\top \mathbf{X}^* \mathbf{u}_i \right),
\end{aligned}
$$

where both (a) and (b) follow from the definition of $r$, and (c) follows since $\sum_{i=1}^n \mathbf{u}_i^\top \mathbf{X}^* \mathbf{u}_i = \mathrm{Tr}(\mathbf{X}^* \sum_{i=1}^n \mathbf{u}_i \mathbf{u}_i^*) = \mathrm{Tr}(\mathbf{X}^* \mathbf{I}) = k$.

Let us denote the projection matrix onto the span of the top $k$ eigenvectors of $-\nabla f(\mathbf{X}^*)$ by $\mathbf{P}^* = \sum_{i=1}^k \mathbf{u}_i \mathbf{u}_i^\top$, and note that $\langle \mathbf{P}^*, -\nabla f(\mathbf{X}^*) \rangle = \sum_{i=1}^k \mu_i$. It follows that

$$
\begin{aligned}
\langle \mathbf{P}^* - \mathbf{X}^*, \nabla f(\mathbf{X}^*) \rangle &= \langle \mathbf{X}^* - \mathbf{P}^*, -\nabla f(\mathbf{X}^*) \rangle \\
&< \sum_{i=1}^{k-1} \mu_i \mathbf{u}_i^\top \mathbf{X}^* \mathbf{u}_i + \mu_k \left( k - \sum_{i=1}^{k-1} \mathbf{u}_i^\top \mathbf{X}^* \mathbf{u}_i \right) - \sum_{i=1}^k \mu_i \\
&= \sum_{i=1}^{k-1} \mu_i \left( \mathbf{u}_i^\top \mathbf{X}^* \mathbf{u}_i - 1 \right) + \mu_k \sum_{i=1}^{k-1} \left( 1 - \mathbf{u}_i^\top \mathbf{X}^* \mathbf{u}_i \right) \\
&= \sum_{i=1}^{k-1} \left( 1 - \mathbf{u}_i^\top \mathbf{X}^* \mathbf{u}_i \right) (\mu_k - \mu_i) \leq 0,
\end{aligned}
$$

where the last inequality follows since for all $i$, $\mathbf{u}_i^\top \mathbf{X}^* \mathbf{u}_i \in [0, 1]$.

Thus, we have that $\mathbf{X}^*$ violates the first-order optimality condition which contradicts that assumption that it is an optimal solution, and thus we have that $\mathbf{X}^*$ must indeed be orthogonal to $\mathbf{u}_{r+1}\mathbf{u}_{r+1}^\top, \ldots, \mathbf{u}_n\mathbf{u}_n^\top$.

An immediate consequence is that the eigenvectors of $\mathbf{X}^*$ which correspond to non-zero eigenvalues must lie in $\mathrm{span}\{\mathbf{u}_1, ..., \mathbf{u}_r\}$ and thus, it must be that $\mathrm{rank}(\mathbf{X}^*) \leq r$.

For the final part of the lemma, in case $r = k$, since for all $\mathbf{X} \in \mathcal{F}_{n,k}$, $\mathrm{rank}(\mathbf{X}) \geq k$, we have that $\mathrm{rank}(\mathbf{X}^*) = k$. In particular, $\mathbf{X}^*$ is a projection matrix, i.e., $\mathbf{X} \in \mathcal{P}_{n,k}$. By the orthogonality result above, it follows that the eigenvectors of $\mathbf{X}^*$ lie in $\mathrm{span}\{\mathbf{u}_1, \ldots, \mathbf{u}_k\}$, which means that $\mathbf{X}^*$ is indeed the projection matrix onto $\mathrm{span}\{\mathbf{u}_1, \ldots, \mathbf{u}_k\}$, as stated in the lemma. Note that when $r = k$, this projection matrix is indeed unique (i.e., the subspace spanned by the top $k$ eigenvectors of $-\nabla f(\mathbf{X}^*)$ is unique). $\qquad \square$

We now prove Theorem 1.

*Proof of Theorem 1.* Let $\mathbf{X}^*$ be an optimal solution to the convex relaxation (4) which satisfies Assumption 1 with some $\delta > 0$. It follows directly from Lemma 6 that $\mathrm{rank}(\mathbf{X}^*) = k$. From Lemma 6 it further follows that $\mathbf{X}^*$ is the unique projection matrix onto the span of top $k$ eigenvectors of $-\nabla f(\mathbf{X}^*)$, i.e., it is the unique matrix in $\mathbf{X} \in \mathcal{P}_{n,k}$ such that $\langle \mathbf{X}, -\nabla f(\mathbf{X}^*) \rangle = \sum_{i=1}^k \mu_i$, where we write the eigen-decomposition of $-\nabla f(\mathbf{X}^*)$ as $-\nabla f(\mathbf{X}^*) = \sum_{i=1}^n \mu_i \mathbf{u}_i \mathbf{u}_i^\top$. From the von Neumann trace inequality it follows that for any matrix $\mathbf{X} \in \mathcal{P}_{n,k}$ it holds that $\langle \mathbf{X}, -\nabla f(\mathbf{X}^*) \rangle \leq \sum_{i=1}^n \lambda_i(\mathbf{X})\mu_i = \sum_{i=1}^k \lambda_i(\mathbf{X})\mu_i = \sum_{i=1}^k \mu_i$. Thus, we have that for all $\mathbf{X} \in \mathcal{P}_{n,k} \setminus \{\mathbf{X}^*\}$ : $\langle \mathbf{X} - \mathbf{X}^*, \nabla f(\mathbf{X}^*) \rangle = \langle \mathbf{X}^* - \mathbf{X}, -\nabla f(\mathbf{X}^*) \rangle > 0$. Since $\mathcal{F}_{n,k} = \mathrm{conv}\{\mathcal{P}_{n,k}\}$, this further implies that for all $\mathbf{X} \in \mathcal{F}_{n,k} \setminus \{\mathbf{X}^*\}$: $\langle \mathbf{X} - \mathbf{X}^*, \nabla f(\mathbf{X}^*) \rangle > 0$. Since $f(\cdot)$ is convex, it further holds that

for all $\mathbf{X} \in \mathcal{F}_{n,k} \setminus \{\mathbf{X}^*\}$: $f(\mathbf{X}^*) - f(\mathbf{X}) \le \langle \mathbf{X}^* - \mathbf{X}, \nabla f(\mathbf{X}^*)\rangle < 0$, and thus, we conclude that $\mathbf{X}^*$ is indeed the unique optimal solution to Problem (4), which also implies that it is the unique optimal solution to Problem (1). $\qquad\square$

### C.3  Proof of Lemma 3

*Proof.* Let us write the eigen-decomposition of the gradient $\nabla f(\mathbf{X}^*)$ as $\nabla f(\mathbf{X}^*) = \sum_{i=1}^{n} \lambda_i \mathbf{u}_i \mathbf{u}_i^\top$. For any $\mathbf{X} \in \mathcal{F}_{n,k}$ it holds that:

$$f(\mathbf{X}) - f(\mathbf{X}^*) \underset{(a)}{\ge} \langle \mathbf{X} - \mathbf{X}^*, \nabla f(\mathbf{X}^*)\rangle \underset{(b)}{=} \sum_{i=1}^{n} \lambda_i \mathbf{u}_i^\top \mathbf{X} \mathbf{u}_i - \sum_{i=n-k+1}^{n} \lambda_i$$

$$\underset{(c)}{\ge} (\lambda_{n-k+1} + \delta) \sum_{i=1}^{n-k} \mathbf{u}_i^\top \mathbf{X} \mathbf{u}_i + \sum_{i=n-k+1}^{n} \lambda_i \mathbf{u}_i^\top \mathbf{X} \mathbf{u}_i - \sum_{i=n-k+1}^{n} \lambda_i$$

$$= (\lambda_{n-k+1} + \delta) \sum_{i=1}^{n-k} \mathbf{u}_i^\top \mathbf{X} \mathbf{u}_i - \sum_{i=n-k+1}^{n} \lambda_i (1 - \mathbf{u}_i^\top \mathbf{X} \mathbf{u}_i)$$

$$\underset{(d)}{\ge} (\lambda_{n-k+1} + \delta) \sum_{i=1}^{n-k} \mathbf{u}_i^\top \mathbf{X} \mathbf{u}_i - \lambda_{n-k+1} \sum_{i=n-k+1}^{n} (1 - \mathbf{u}_i^\top \mathbf{X} \mathbf{u}_i)$$

$$= \lambda_{n-k+1} \sum_{i=1}^{n} \mathbf{u}_i^\top \mathbf{X} \mathbf{u}_i - k\lambda_{n-k+1} + \delta \sum_{i=1}^{n-k} \mathbf{u}_i^\top \mathbf{X} \mathbf{u}_i, \tag{18}$$

where (a) follows from the convexity of $f(\mathbf{X})$, (b) follows since according Lemma 6 $\mathbf{X}^* = \sum_{i=n-k+1}^{n} \mathbf{u}_i \mathbf{u}_i^\top$ and so, $\langle \mathbf{X}^*, \nabla f(\mathbf{X}^*)\rangle = \sum_{i=n-k+1}^{n} \lambda_i$, (c) follows from Assumption 1, and (d) follows since $\mathbf{X} \preceq \mathbf{I}$, which implies that $\mathbf{u}_i^\top \mathbf{X} \mathbf{u}_i \le \mathbf{u}_i^\top \mathbf{u}_i = 1$.

Using $\sum_{i=1}^{n} \mathbf{u}_i^\top \mathbf{X} \mathbf{u}_i = \mathrm{Tr}(\mathbf{X} \sum_{i=1}^{n} \mathbf{u}_i \mathbf{u}_i^\top) = \mathrm{Tr}(\mathbf{X}\mathbf{I}) = k$ and Eq. (18), we have,

$$f(\mathbf{X}) - f(\mathbf{X}^*) \ge \delta \sum_{i=1}^{n-k} \mathbf{u}_i^\top \mathbf{X} \mathbf{u}_i = \delta \left( k - \sum_{i=n-k+1}^{n} \mathbf{u}_i^\top \mathbf{X} \mathbf{u}_i \right). \tag{19}$$

Also, using again the fact that $\mathbf{X}^* = \sum_{i=n-k+1}^{n} \mathbf{u}_i \mathbf{u}_i^\top$, we have that,

$$\|\mathbf{X} - \mathbf{X}^*\|_F^2 = \|\mathbf{X}\|_F^2 + \|\mathbf{X}^*\|_F^2 - 2 \sum_{i=n-k+1}^{n} \mathbf{u}_i^\top \mathbf{X} \mathbf{u}_i \le 2 \left( k - \sum_{i=n-k+1}^{n} \mathbf{u}_i^\top \mathbf{X} \mathbf{u}_i \right), \tag{20}$$

where the last inequality follows since for any $\mathbf{X} \in \mathcal{F}_{n,k}$ it holds that $\|\mathbf{X}\|_F^2 = \sum_{i=1}^{n} \lambda_i^2(\mathbf{X}) \le \sum_{i=1}^{n} \lambda_i(\mathbf{X}) = k$. Combining Eq. (19) and (20) we finally have that,

$$f(\mathbf{X}) - f(\mathbf{X}^*) \ge \frac{\delta}{2} \|\mathbf{X} - \mathbf{X}^*\|_F^2.$$

$\qquad\square$

## D  Projected Gradient Descent Analysis

In this section we turn to analyze the local convergence of the projected gradient method w.r.t. the sets $\mathcal{P}_{n,k}$ and $\mathcal{F}_{n,k}$, and to prove Theorems 4 and 6.

We first provide the proof of Lemma 2 which is fairly simple, and then prove a more general version of the lemma, which in particular allows to relax Assumption 1.

*Proof of Lemma 2.* Denote $\mathbf{Y}^* = \mathbf{X}^* - \eta \nabla f(\mathbf{X}^*)$ and denote the eigenvalues of $\mathbf{Y}^*$ in non-increasing order $\sigma_i = \lambda_i(\mathbf{Y}^*), i = 1, \ldots, n$. Denote also $\mathbf{Y} = \mathbf{X} - \eta \nabla f(\mathbf{X})$ with its eigenvalues $\gamma_i = \lambda_i(\mathbf{Y}), i = 1, \ldots, n$. Let us write the eigen-decomposition of $-\nabla f(\mathbf{X}^*)$ as

$-\nabla f(\mathbf{X}^*) = \sum_{i=1}^{n} \mu_i \mathbf{u}_i \mathbf{u}_i^\top$. From Lemma 6, we have that under Assumption 1, it holds that $\mathbf{X}^* = \sum_{i=1}^{k} \mathbf{u}_i \mathbf{u}_i^\top$. Thus, we can deduce that

$$\sigma_i = \begin{cases} 1 + \eta\mu_i & \text{if } i \in \{1, ..., k\}; \\ \eta\mu_i & \text{else.} \end{cases} \tag{21}$$

From Lemma 1 we have that $\text{rank}(\Pi_{\mathcal{F}_{n,k}}(\mathbf{Y})) = k$ if and only if $\sum_{i=1}^{k} \min(\gamma_i - \gamma_{k+1}, 1) \geq k$. Thus, a sufficient condition so that $\text{rank}(\Pi_{\mathcal{F}_{n,k}}(\mathbf{Y})) = k$ is,

$$\gamma_k - \gamma_{k+1} \geq 1. \tag{22}$$

By Weyl's inequality for the eigenvalues and Eq. (21) we have,

$$\begin{aligned} \gamma_k - \gamma_{k+1} &= (\sigma_k - \sigma_{k+1}) + (\gamma_k - \sigma_k) + (\sigma_{k+1} - \gamma_{k+1}) \\ &\geq 1 + \eta(\mu_k - \mu_{k+1}) - 2\|\mathbf{Y} - \mathbf{Y}^*\|_F \\ &= 1 + \eta(\mu_k - \mu_{k+1}) - 2\|\mathbf{X} - \mathbf{X}^* - \eta\nabla f(\mathbf{X}) + \eta\nabla f(\mathbf{X}^*)\|_F \\ &\geq 1 + \eta(\mu_k - \mu_{k+1}) - 2(1 + \eta\beta)\|\mathbf{X} - \mathbf{X}^*\|_F. \end{aligned}$$

Thus, we see that a sufficient condition so that (22) holds, is that $\mathbf{X}$ satisfies

$$\|\mathbf{X} - \mathbf{X}^*\|_F \leq \frac{\eta\delta}{2(1 + \eta\beta)} \leq \frac{\eta(\mu_k - \mu_{k+1})}{2(1 + \eta\beta)},$$

and so the lemma follows. $\square$

The following lemma generalizes Lemma 2 and offers a natural trade-off between the rank of the projected gradient mapping and the size of the ball around an optimal solution $\mathbf{X}^*$ in which it is guaranteed to be upper-bounded.

**Lemma 7.** *Let $\mathbf{X}^* \in \mathcal{F}_{n,k}$ be an optimal solution to Problem (4), and let $\mu_1 \geq \mu_2 \geq \ldots \mu_n$ denote the eigenvalues of $-\nabla f(\mathbf{X}^*)$. Let $r$ be the smallest integer such that $r \geq k$ and $\mu_k > \mu_{r+1}$. Fix some $\eta > 0$. For any $\mathbf{X} \in \mathcal{F}_{n,k}$ which satisfies*

$$\|\mathbf{X} - \mathbf{X}^*\|_F \leq \frac{\eta(\mu_k - \mu_{r+1})}{2(1 + \eta\beta)}, \tag{23}$$

*it holds that $\text{rank}(\Pi_{\mathcal{F}_k}(\mathbf{X} - \eta\nabla f(\mathbf{X}))) \leq r$.*
*More generally, for any $r' \in \{r, r+1, ..., n-1\}$ and for any $\eta > 0$, if $\mathbf{X} \in \mathcal{F}_{n,k}$ satisfies*

$$\|\mathbf{X} - \mathbf{X}^*\|_F \leq \frac{\eta(\mu_k - \mu_{r'+1})}{2(1 + \eta\beta)}, \tag{24}$$

*then $\text{rank}(\Pi_{\mathcal{F}_k}(\mathbf{X} - \eta\nabla f(\mathbf{X}))) \leq r'$.*

*Proof.* From Lemma 6 we have that $r^* := \text{rank}(\mathbf{X}^*) \leq r$. Denote $\mathbf{Y}^* = \mathbf{X}^* - \eta\nabla f(\mathbf{X}^*)$ and denote the eigenvalues of $\mathbf{Y}^*$ as $\sigma_i = \lambda_i(\mathbf{Y}^*), i = 1, \ldots, n$. Denote also $\mathbf{Y} = \mathbf{X} - \eta\nabla f(\mathbf{X})$ with its eigenvalues $\gamma_i = \lambda_i(\mathbf{Y}), i = 1, \ldots, n$.
From the min-max principle for the eigenvalues, letting $\mathcal{V} \subseteq \mathbb{R}^n$ denote some subspace of $\mathbb{R}^n$, we have that for any $i \in \{1, ..., r\}$,

$$\sigma_i = \min_{\mathcal{V}:\dim(\mathcal{V})=n-i+1} \max_{\mathbf{v}\in\mathcal{V}:\|\mathbf{v}\|=1} \mathbf{v}^\top(\mathbf{X}^* + \eta(-\nabla f(\mathbf{X}^*)))\mathbf{v}. \tag{25}$$

Let us write the eigen-decomposition of $-\nabla f(\mathbf{X}^*)$ as $-\nabla f(\mathbf{X}^*) = \sum_{i=1}^{n} \mu_i \mathbf{u}_i \mathbf{u}_i^\top$. Note that in Eq. (25) we minimize over all the subspaces $\mathcal{V}$ of dimension $n - i + 1$, $i \leq r$, and so,

$$\mathcal{V} \cap \text{span}\{\mathbf{u}_1, ..., \mathbf{u}_r\} \neq \emptyset, \tag{26}$$

otherwise the direct sum $\mathcal{V} \oplus \text{span}\{\mathbf{u}_1, ..., \mathbf{u}_r\} \subseteq \mathbb{R}^n$ would have dimension $n - i + 1 + r > n$.

Any unit vector $\mathbf{v} \in \mathcal{V}$ can be written as $\mathbf{v} = a\mathbf{u} + b\mathbf{w}$ such that $\mathbf{u} \in \text{span}\{\mathbf{u}_1, ..., \mathbf{u}_r\}, \|\mathbf{u}\| = 1$, $\mathbf{w} \in \text{span}\{\mathbf{u}_{r+1}, ..., \mathbf{u}_n\}, \|\mathbf{w}\| = 1$, and $a^2 + b^2 = 1$. Thus, for any such unit vector $\mathbf{v}$, using Lemma 6, we have that,

$$\mathbf{v}^\top(\mathbf{X}^* + \eta(-\nabla f(\mathbf{X}^*)))\mathbf{v} = a^2\mathbf{u}^\top\mathbf{X}^*\mathbf{u} + a^2\eta\mathbf{u}^\top(-\nabla f(\mathbf{X}^*))\mathbf{u} + b^2\eta\mathbf{w}^\top(-\nabla f(\mathbf{X}^*))\mathbf{w}. \tag{27}$$

Note that

$$\mathbf{u}^\top(-\nabla f(\mathbf{X}^*))\mathbf{u} \geq \mu_r > \mu_{r+1} \geq \mathbf{w}^\top(-\nabla f(\mathbf{X}^*))\mathbf{w}. \tag{28}$$

This implies that the inner maximum in (25) can only be obtained by vectors in $\mathcal{V} \cap \mathrm{span}\{\mathbf{u}_1, ..., \mathbf{u}_r\}$ (note (26) guarantees such vectors exist). Thus, plugging this observation into (25) we have that for any $i \in \{1, \ldots, r\}$,

$$\sigma_i = \min_{\mathcal{V}:\dim(\mathcal{V})=n-i+1} \max_{\mathbf{v}\in\mathcal{V}\cap\mathrm{span}\{\mathbf{u}_1,...,\mathbf{u}_r\},\|\mathbf{v}\|=1} \mathbf{v}^\top(\mathbf{X}^* + \eta(-\nabla f(\mathbf{X}^*)))\mathbf{v}$$

$$\geq \min_{\mathcal{V}:\dim(\mathcal{V})=n-i+1} \max_{\mathbf{v}\in\mathcal{V}\cap\mathrm{span}\{\mathbf{u}_1,...,\mathbf{u}_r\},\|\mathbf{v}\|=1} \mathbf{v}^\top\mathbf{X}^*\mathbf{v} + \eta\mu_r$$

$$\underset{(a)}{=} \min_{\mathcal{V}:\dim(\mathcal{V})=n-i+1} \max_{\mathbf{v}\in\mathcal{V},\|\mathbf{v}\|=1} \mathbf{v}^\top\mathbf{X}^*\mathbf{v} + \eta\mu_r \underset{(b)}{=} \lambda_i(\mathbf{X}^*) + \eta\mu_k, \tag{29}$$

where (a) follows from the orthogonality of $\mathbf{X}^*$ to $\mathbf{u}_{r+1}\mathbf{u}_{r+1}^\top, ..., \mathbf{u}_n\mathbf{u}_n^\top$ (see Lemma 6), and (b) follows from the min-max principle for the eigenvalues, and since by definition $\mu_r = \mu_k$.

Using the max-min principle for the eigenvalues, we can write for any $j \in \{r+1, ..., n\}$,

$$\sigma_j = \max_{\mathcal{V}:\dim(\mathcal{V})=j} \min_{\mathbf{v}\in\mathcal{V}:\|\mathbf{v}\|=1} \mathbf{v}^\top(\mathbf{X}^* + \eta(-\nabla f(\mathbf{X}^*)))\mathbf{v}. \tag{30}$$

This time we maximize over all subspaces of dimension $j$, $j \geq r+1$. Thus, it must hold that for each such subspace $\mathcal{V}$,

$$\mathcal{V} \cap \mathrm{span}\{\mathbf{u}_{r+1}, ..., \mathbf{u}_n\} \neq \emptyset,$$

otherwise the direct sum $\mathcal{V} \oplus \mathrm{span}\{\mathbf{u}_{r+1}, ..., \mathbf{u}_n\} \subseteq \mathbb{R}^n$ would have dimension $j + n - r > n$. Thus, using (27) and (28), we have that the inner minimum in (30) is obtained by vectors in $\mathcal{V} \cap \mathrm{span}\{\mathbf{u}_{r+1}, \ldots, \mathbf{u}_n\}$, which is not an empty set. Using this observation we have that for any $j \in \{r+1, \ldots, n\}$,

$$\sigma_j = \max_{\mathcal{V}:\dim(\mathcal{V})=j} \min_{\mathbf{v}\in\mathcal{V}\cap\mathrm{span}\{\mathbf{u}_{r+1},...,\mathbf{u}_n\},\|\mathbf{v}\|=1} \mathbf{v}^\top(\mathbf{X}^* + \eta(-\nabla f(\mathbf{X}^*)))\mathbf{v}$$

$$\underset{(a)}{=} \max_{\mathcal{V}:\dim(\mathcal{V})=j} \min_{\mathbf{v}\in\mathcal{V}\cap\mathrm{span}\{\mathbf{u}_{r+1},...,\mathbf{u}_n\},\|\mathbf{v}\|=1} \mathbf{v}^\top(\eta(-\nabla f(\mathbf{X}^*)))\mathbf{v}$$

$$\underset{(b)}{=} \max_{\mathcal{V}:\dim(\mathcal{V})=j} \min_{\mathbf{v}\in\mathcal{V},\|\mathbf{v}\|=1} \mathbf{v}^\top(\eta(-\nabla f(\mathbf{X}^*)))\mathbf{v} = \eta\mu_j, \tag{31}$$

where (a) follows since $\mathbf{X}^*$ is orthogonal to $\mathbf{u}_{r+1}\mathbf{u}_{r+1}^\top, \ldots, \mathbf{u}_n\mathbf{u}_n^\top$ (see Lemma 6), and (b) follows since by the eigen-decomposition of $-\nabla f(\mathbf{X}^*)$, restricting $\mathbf{v}$ to the intersection $\mathcal{V} \cap \mathrm{span}\{\mathbf{u}_{r+1}, ..., \mathbf{u}_n\}$ does not increase the inner minimum.

From Lemma 1 we have the sufficient condition so that $\mathrm{rank}(\Pi_{\mathcal{F}_{n,k}}(\mathbf{Y})) \leq r$:

$$\sum_{i=1}^r \min(\gamma_i - \gamma_{r+1}, 1) \geq k \implies \mathrm{rank}(\Pi_{\mathcal{F}_{n,k}}(\mathbf{Y})) \leq r. \tag{32}$$

By Weyl's inequality we have that for any $i \in \{1, \ldots, r\}$,

$$\gamma_i - \gamma_{r+1} \geq \sigma_i - \sigma_{r+1} - 2\|\mathbf{Y} - \mathbf{Y}^*\|_F$$

$$= \sigma_i - \sigma_{r+1} - 2\|\mathbf{X} - \eta\nabla f(\mathbf{X}) - \mathbf{X}^* + \eta\nabla f(\mathbf{X}^*)\|_F$$

$$\geq \sigma_i - \sigma_{r+1} - 2(1+\eta\beta)\|\mathbf{X} - \mathbf{X}^*\|_F. \tag{33}$$

Thus, we have that

$$\sum_{i=1}^r \min(\gamma_i - \gamma_{r+1}, 1) \underset{(a)}{\geq} \sum_{i=1}^r \min(\sigma_i - \sigma_{r+1} - 2(1+\eta\beta)\|\mathbf{X} - \mathbf{X}^*\|_F, 1)$$

$$\underset{(b)}{\geq} \sum_{i=1}^r \min(\lambda_i(\mathbf{X}^*) + \eta(\mu_i - \mu_{r+1}) - 2(1+\eta\beta)\|\mathbf{X} - \mathbf{X}^*\|_F, 1)$$

$$\geq \sum_{i=1}^r \min(\lambda_i(\mathbf{X}^*) + \eta(\mu_r - \mu_{r+1}) - 2(1+\eta\beta)\|\mathbf{X} - \mathbf{X}^*\|_F, 1), \tag{34}$$

where (a) follows from (33), and (b) follows from (29) and (31).

Thus, we indeed see that if

$$\|\mathbf{X} - \mathbf{X}^*\|_F \leq \frac{\eta(\mu_r - \mu_{r+1})}{2(1 + \eta\beta)},$$

then $\sum_{i=1}^{r} \min(\gamma_i - \gamma_{r+1}, 1) \geq \sum_{i=1}^{r^*} \lambda_i(\mathbf{X}^*) = k$, which by (32) implies that $\text{rank}(\Pi_{\mathcal{F}_{n,k}}(\mathbf{Y})) \leq r$, as needed.

For the second part of the lemma let us fix some $r' \in \{r, ..., n-1\}$. If we have that

$$\|\mathbf{X} - \mathbf{X}^*\|_F \leq \frac{\eta(\mu_r - \mu_{r'+1})}{2(1 + \eta\beta)}, \tag{35}$$

then similarly to (34), we will have that,

$$
\begin{aligned}
\sum_{i=1}^{r'} \min(\gamma_i - \gamma_{r'+1}, 1) &\geq \sum_{i=1}^{r} \min(\gamma_i - \gamma_{r'+1}, 1) \\
&\underset{(a)}{\geq} \sum_{i=1}^{r} \min(\sigma_i - \sigma_{r'+1} - 2(1+\eta\beta)\|\mathbf{X} - \mathbf{X}^*\|_F, 1) \\
&\underset{(b)}{\geq} \sum_{i=1}^{r} \min(\lambda_i(\mathbf{X}^*) + \eta(\mu_i - \mu_{r'+1}) - 2(1+\eta\beta)\|\mathbf{X} - \mathbf{X}^*\|_F, 1) \\
&\geq \sum_{i=1}^{r} \min(\lambda_i(\mathbf{X}^*) + \eta(\mu_r - \mu_{r'+1}) - 2(1+\eta\beta)\|\mathbf{X} - \mathbf{X}^*\|_F, 1) \\
&\underset{(c)}{\geq} \sum_{i=1}^{r} \min(\lambda_i(\mathbf{X}^*), 1) = \sum_{i=1}^{r^*} \lambda_i(\mathbf{X}^*) = k,
\end{aligned}
$$

where (a) follows from the same reasoning as (33), (b) follows from (29) and (31), and (c) follows from (35).

Thus, from Lemma 1 we have that (35) indeed implies that $\text{rank}(\Pi_{\mathcal{F}_{n,k}}(\mathbf{Y})) \leq r'$, which proves the second part of the lemma. $\qquad\square$

We can now easily prove Theorems 4 and 6 by proving the following unifying theorem.

**Theorem 7.** *Let $\{\mathbf{X}_t\}_{t\geq 1}$ be a sequence produced by the projected gradient dynamics w.r.t. the convex Problem (4) with a fixed step-size $\eta \in (0, 1/\beta]$:*

$$\mathbf{X}_{t+1} = \Pi_{\mathcal{F}_{n,k}}(\mathbf{X}_t - \eta\nabla f(\mathbf{X}_t)).$$

*Fix some optimal solution $\mathbf{X}^*$ and let $\mu_1 \geq \mu_2 \geq ...\mu_n$ denote the eigenvalues of $-\nabla f(\mathbf{X}^*)$. Let $r$ be the smallest integer such that $r \geq k$ and $\mu_r - \mu_{r+1} > 0$. If the initialization $\mathbf{X}_1 \in \mathcal{F}_{n,k}$ satisfies $\|\mathbf{X}_1 - \mathbf{X}^*\|_F \leq \frac{\eta(\mu_k - \mu_{r+1})}{2(1 + \eta\beta)}$, then for all $t \geq 1$, $rank(\mathbf{X}_{t+1}) \leq r$.*

*More generally, for every $r' \in \{r, \ldots, n\}$, if $\|\mathbf{X}_1 - \mathbf{X}^*\|_F \leq \frac{\eta(\mu_k - \mu_{r'+1})}{2(1 + \eta\beta)}$, then for all $t \geq 1$, $rank(\mathbf{X}_{t+1}) \leq r'$.*

*In particular, if $r = k$, i.e., Assumption 1 holds with some $\delta > 0$, and $\|\mathbf{X}_1 - \mathbf{X}^*\|_F \leq \frac{\delta}{4\beta}$, setting $\eta = 1/\beta$ guarantees that for all $t \geq 1$, $rank(\mathbf{X}_{t+1}) = k$, and the sequence $\{\mathbf{X}_t\}_{t\geq 1}$ converges linearly with rate:*

$$\forall t \geq 1: \quad f(\mathbf{X}_t) - f(\mathbf{X}^*) \leq (f(\mathbf{X}_1) - f(\mathbf{X}^*)) \exp\left(-\Theta\left(\delta/\beta\right)(t-1)\right).$$

*Proof.* It is a well known fact that the distances of the iterates generated by the projected gradient method with step-size $\eta \in (0, 1/\beta]$ to any optimal solution are monotone non-increasing, i.e., the

sequence $\{\|\mathbf{X}_t - \mathbf{X}^*\|_F\}_{t \geq 1}$ is monotone non-increasing, see for instance [3]. Thus, all results of the theorem regarding the rank of the iterates $\mathbf{X}_t, t \geq 1$, follow immediately from this observation, the initialization conditions listed in the theorem, and Lemma 7.

The linear convergence rate under Assumption 1 follows from the quadratic growth result — Lemma 3, and the known linear convergence rate of the projected gradient method for smooth functions that satisfy the quadratic growth property, see for instance [19]. □

# E  Details Missing from Section 3.2

We prove two auxiliary lemmas and then prove Theorem 3.

**Lemma 8.** *Let $\mathbf{M} \in \mathbb{S}^n$, and let $\mathbf{X} \in \mathcal{P}_{n,k}$ be the projection matrix onto the span of the top $k$ eigenvectors of $\mathbf{M}$. Then, for any $\mathbf{Z} \in \mathcal{P}_{n,k}$ it holds that,*

$$\langle \mathbf{X} - \mathbf{Z}, \mathbf{M} \rangle \leq \|\mathbf{Z} - \mathbf{X}\|_F^2 \|\mathbf{M}\|_2.$$

*Proof.* Let us denote by $\mathbf{X}_\perp$ the projection matrix onto the orthogonal subspace, i.e., $\mathbf{X}_\perp = \mathbf{I} - \mathbf{X}$. It holds that,

$$
\begin{aligned}
\langle \mathbf{X} - \mathbf{Z}, \mathbf{M} \rangle &= \langle \mathbf{X} - \mathbf{Z}, \mathbf{X}\mathbf{M} \rangle + \langle \mathbf{X} - \mathbf{Z}, \mathbf{X}_\perp \mathbf{M} \rangle \\
&= \langle \mathbf{X} - \mathbf{Z}, \mathbf{X}\mathbf{M} \rangle - \langle \mathbf{Z}, \mathbf{X}_\perp \mathbf{M} \rangle.
\end{aligned}
\tag{36}
$$

We consider each of the two terms on the RHS separately.

$$
\begin{aligned}
\langle \mathbf{X} - \mathbf{Z}, \mathbf{X}\mathbf{M} \rangle &= \mathrm{Tr}((\mathbf{X} - \mathbf{Z})\mathbf{X}\mathbf{M}) = \mathrm{Tr}(\mathbf{X}(\mathbf{X} - \mathbf{Z})\mathbf{X}\mathbf{M}) \\
&\underset{(a)}{\leq} \mathrm{Tr}(\mathbf{X}(\mathbf{X} - \mathbf{Z})\mathbf{X}) \cdot \lambda_1(\mathbf{M}) \\
&= \langle \mathbf{X} - \mathbf{Z}, \mathbf{X} \rangle \cdot \lambda_1(\mathbf{M}) = (k - \langle \mathbf{Z}, \mathbf{X} \rangle) \cdot \lambda_1(\mathbf{M}) \\
&\leq \frac{1}{2}\|\mathbf{Z} - \mathbf{X}\|_F^2 \cdot \|\mathbf{M}\|_2,
\end{aligned}
\tag{37}
$$

where $(a)$ holds since $\mathbf{X}(\mathbf{X} - \mathbf{Z})\mathbf{X}$ is positive semidefinite.

$$
\begin{aligned}
\langle \mathbf{Z}, \mathbf{X}_\perp \mathbf{M} \rangle &= \mathrm{Tr}(\mathbf{Z}\mathbf{X}_\perp \mathbf{M}) = \mathrm{Tr}(\mathbf{X}_\perp \mathbf{Z}\mathbf{X}_\perp \mathbf{M}) \\
&\underset{(b)}{\geq} \mathrm{Tr}(\mathbf{X}_\perp \mathbf{Z}\mathbf{X}_\perp) \cdot \lambda_n(\mathbf{M}) = \langle \mathbf{Z}, \mathbf{X}_\perp \rangle \cdot \lambda_n(\mathbf{M}) \\
&= \langle \mathbf{Z}, \mathbf{I} - \mathbf{X} \rangle \cdot \lambda_n(\mathbf{M}) = (k - \langle \mathbf{Z}, \mathbf{X} \rangle) \cdot \lambda_n(\mathbf{M}) \\
&\geq -\frac{1}{2}\|\mathbf{Z} - \mathbf{X}^*\|_F^2 \cdot \|\mathbf{M}\|_2,
\end{aligned}
\tag{38}
$$

where $(b)$ holds since $\mathbf{X}_\perp \mathbf{Z}\mathbf{X}_\perp$ is positive semidefinite.

The lemma follows from plugging (37) and (38) into (36). □

**Lemma 9.** *Fix some $t \geq 1$ and suppose $\eta < 1/\beta$. Then it holds that,*

$$\|\mathbf{X}_{t+1} - \mathbf{Y}_t\|_F^2 \leq \frac{\eta}{1 - \eta\beta} \left( f(\mathbf{Y}_t) - f(\mathbf{X}_{t+1}) \right).$$

*Proof.* Define the following function

$$\phi(\mathbf{Z}) := \langle \mathbf{Z}, \nabla f(\mathbf{Y}_t) \rangle + \frac{1}{2\eta}\|\mathbf{Z} - \mathbf{Y}_t\|_F^2,$$

and note that it is $1/\eta$ strongly convex, and that by definition, $\mathbf{X}_{t+1}$ is its minimizer over $\mathcal{F}_{n,k}$. Thus,

$$
\begin{aligned}
\|\mathbf{X}_{t+1} - \mathbf{Y}_t\|_F^2 &\leq 2\eta \left( \phi(\mathbf{Y}_t) - \phi(\mathbf{X}_{t+1}) \right) \\
&= 2\eta\langle \mathbf{Y}_t - \mathbf{X}_{t+1}, \nabla f(\mathbf{Y}_t) \rangle - \|\mathbf{X}_{t+1} - \mathbf{Y}_t\|_F^2.
\end{aligned}
$$

Rearranging we get,

$$\|\mathbf{X}_{t+1} - \mathbf{Y}_t\|_F^2 \leq \eta\langle \mathbf{Y}_t - \mathbf{X}_{t+1}, \nabla f(\mathbf{Y}_t)\rangle$$
$$= \eta\langle \mathbf{Y}_t - \mathbf{X}_{t+1}, \nabla f(\mathbf{X}_{t+1})\rangle + \eta\langle \mathbf{Y}_t - \mathbf{X}_{t+1}, \nabla f(\mathbf{Y}_t) - \nabla f(\mathbf{X}_{t+1})\rangle$$
$$\underset{(a)}{\leq} \eta\langle \mathbf{Y}_t - \mathbf{X}_{t+1}, \nabla f(\mathbf{X}_{t+1})\rangle + \eta\beta\|\mathbf{X}_{t+1} - \mathbf{Y}_t\|_F^2$$
$$\underset{(b)}{\leq} \eta\left(f(\mathbf{Y}_t) - f(\mathbf{X}_{t+1})\right) + \eta\beta\|\mathbf{X}_{t+1} - \mathbf{Y}_t\|_F^2,$$

where (a) follows from the $\beta$-smoothness of $f(\cdot)$, and (b) follows from the convexity of $f(\cdot)$. Rearranging, we get the lemma. $\qquad\square$

*Proof of Theorem 3.* The theorem follows from Lemma 5, it only remains to prove that the necessary conditions hold for all $t \geq 1$, i.e., that for all $t \geq 1$, it holds that $\|\mathbf{X}_{t+1} - \mathbf{Y}_t\|_F \leq 1$, and $\mathbf{X}_{t+1} \in \mathcal{P}_{n,k}$, i.e., rank$(\mathbf{X}_{t+1}) = k$. The proof is by induction. For the base case $t = 1$, we first note that using Lemma 9 we have that,

$$\|\mathbf{X}_2 - \mathbf{Y}_1\|_F^2 \leq \frac{\eta}{1 - \eta\beta}\left(f(\mathbf{Y}_1) - f(\mathbf{X}_2)\right) \leq \frac{\eta}{1 - \eta\beta}\left(f(\mathbf{Y}_1) - f(\mathbf{X}^*)\right)$$
$$\underset{(a)}{\leq} \frac{\eta}{1 - \eta\beta}\left(\langle \mathbf{X}^* - \mathbf{Y}_1, -\nabla f(\mathbf{X}^*)\rangle + \frac{\beta}{2}\|\mathbf{Y}_1 - \mathbf{X}^*\|_F^2\right)$$
$$\underset{(b)}{\leq} \frac{\eta}{1 - \eta\beta}\left(G + \frac{\beta}{2}\right)\|\mathbf{Y}_1 - \mathbf{X}^*\|_F^2, \tag{39}$$

where (a) follows from the $\beta$-smoothness of $f(\cdot)$, and (b) follows from Lemma 8 and recalling that under Assumption 1, $\mathbf{X}^*$ is the projection matrix onto the span of the top $k$ eigenvectors of $-\nabla f(\mathbf{X}^*)$ (see Lemma 6).

Note that for $\eta \in (0, \beta)$, $\frac{\eta}{1-\eta\beta}\left(G + \frac{\beta}{2}\right) \leq 1 \iff \eta\left(G + \frac{3\beta}{2}\right) \leq 1$, which clearly holds for our choice of step-size $\eta = \frac{1}{5\max\{\beta,G\}}$.

Thus, noting that for our choice of step-size and initialization assumption it holds that $\|\mathbf{Y}_1 - \mathbf{X}^*\|_F \leq 1$, we indeed have that $\|\mathbf{X}_2 - \mathbf{X}_1\|_F \leq 1$. Also, combining the initialization condition for $\mathbf{Y}_1$ listed in the theorem, together with Lemma 2, immediately implies that rank$(\mathbf{X}_2) = k$. Thus, the base case $t = 1$ of the induction holds.

Suppose now the induction holds for all $i \in \{1, \ldots, t-1\}$, for some $t \geq 2$, and we will prove it for $t$. Using Lemma 9 we have that,

$$\|\mathbf{X}_{t+1} - \mathbf{Y}_t\|_F^2 \leq \frac{\eta}{1 - \eta\beta}\left(f(\mathbf{Y}_t) - f(\mathbf{X}_{t+1})\right) \leq \frac{\eta}{1 - \eta\beta}\left(f(\mathbf{Y}_t) - f(\mathbf{X}^*)\right)$$
$$\underset{(a)}{\leq} \frac{\eta}{1 - \eta\beta}\left(f(\mathbf{Y}_1) - f(\mathbf{X}^*)\right) \underset{(b)}{\leq} \frac{\eta}{1 - \eta\beta}\left(G + \frac{\beta}{2}\right)\|\mathbf{Y}_1 - \mathbf{X}^*\|_F^2, \tag{40}$$

where (a) follows by using the induction hypothesis for all $i \leq t$ together with Lemma 5, which guarantees that $f(\mathbf{Y}_t) \leq f(\mathbf{Y}_1)$, and (b) follows from the same steps as in (39).

Since we have already established that the RHS of (40) is upper-bounded by 1 in the base case of the induction, it follows that $\|\mathbf{X}_{t+1} - \mathbf{Y}_t\|_F \leq 1$.

Using the quadratic growth of $f(\cdot)$ (Lemma 3) we have that,

$$\|\mathbf{Y}_t - \mathbf{X}^*\|_F^2 \leq \frac{2}{\delta}\left(f(\mathbf{Y}_t) - f(\mathbf{X}^*)\right) \underset{(a)}{\leq} \frac{2}{\delta}\left(f(\mathbf{Y}_1) - f(\mathbf{X}^*)\right)$$
$$\underset{(b)}{\leq} \frac{2}{\delta}\frac{\eta}{1 - \eta\beta}\left(G + \frac{\beta}{2}\right)\|\mathbf{Y}_1 - \mathbf{X}^*\|_F^2,$$

where again, (a) follows by using the induction hypothesis for all $i \leq t$ together with Lemma 5 which guarantees that $f(\mathbf{Y}_t) \leq f(\mathbf{Y}_1)$, and (b) follows from (39).

Thus, using the fact that for our choice of step-size it holds that $\frac{\eta}{1-\eta\beta}\left(G + \frac{\beta}{2}\right) \leq 1$, using the initalization assumption on $\mathbf{Y}_1$, and invoking Lemma 2, it follows that indeed rank$(\mathbf{X}_{t+1}) = k$, i.e., $\mathbf{X}_{t+1} \in \mathcal{P}_{n,k}$, and thus the induction holds for step $t$ as well. $\qquad\square$

# F    Frank-Wolfe Analysis

In this section we prove Theorem 5. Our analysis extends the one in [8] which only considered the case $k = 1$.

We begin with a lemma, whose proof is similar to the arguments used in the proof of Lemma 3, which will be essential to proving the local linear convergence of Frank-Wolfe under Assumption 1.

**Lemma 10.** *Let* $\mathbf{X} \in \mathcal{F}_{n,k}$ *and assume that* $\lambda_{n-k}(\nabla f(\mathbf{X})) - \lambda_{n-k+1}(\nabla f(\mathbf{X})) \geq \delta_{\mathbf{X}} > 0$. *Then, for* $\mathbf{V} \in \arg\min_{\mathbf{P} \in \mathcal{P}_{n,k}} \langle \mathbf{P}, \nabla f(\mathbf{X}) \rangle$ *it holds that,*

$$\langle \mathbf{X} - \mathbf{V}, \nabla f(\mathbf{X}) \rangle \geq \frac{\delta_{\mathbf{X}}}{2} \|\mathbf{X} - \mathbf{V}\|_F^2.$$

*Proof.* The proof follows the same lines as the proof of Lemma 3, but replacing $\mathbf{X}^*$ with $\mathbf{X}$, and noting that $\mathbf{V}$ is the (unique) projection matrix onto the span of the top $k$ eigenvectors of $-\nabla f(\mathbf{X})$, similarly to the use of $\mathbf{X}^*$ as the projection matrix onto the span of the top $k$ eigenvectors of $-\nabla f(\mathbf{X}^*)$ in the proof of Lemma 3. $\square$

---

**Algorithm 2** Frank-Wolfe with line-search for the Fantope
___

1: $\mathbf{X}_1 \leftarrow$ arbitrary point in $\mathcal{F}_{n,k}$
2: **for** $t = 1...$ **do**
3: $\qquad \mathbf{V}_t \leftarrow \arg\min_{\mathbf{V} \in \mathcal{P}_{n,k}} \langle \mathbf{V}, \nabla f(\mathbf{X}_t) \rangle$
4: $\qquad$ Choose step size $\eta_t \in [0, 1]$ using one of the two options:
5: $\qquad$ First Option : $\eta_t \leftarrow \arg\min_{\eta \in [0,1]} f((1 - \eta)\mathbf{X}_t + \eta \mathbf{V}_t)$
6: $\qquad$ Second Option : $\eta_t \leftarrow \arg\min_{\eta \in [0,1]} f(\mathbf{X}_t) + \eta \langle \mathbf{V}_t - \mathbf{X}_t, \nabla f(\mathbf{X}_t) \rangle + \frac{\eta^2 \beta}{2} \|\mathbf{V}_t - \mathbf{X}_t\|_F^2$
7: $\qquad \mathbf{X}_{t+1} \leftarrow (1 - \eta_t)\mathbf{X}_t + \eta_t \mathbf{V}_t$
8: **end for**

---

**Theorem 8** (Formal version of Theorem 5). *Let* $\{\mathbf{X}_t\}_{t \geq 1}$ *be a sequence produced by Algorithm 2 and denote* $\forall t \geq 1$, $h_t := f(\mathbf{X}_t) - f(\mathbf{X}^*)$. *Then,*

$$\forall t \geq 1 : \quad h_t = O(k\beta/t). \tag{41}$$

*In addition, if Assumption 1 holds with parameter* $\delta > 0$, *then there exists* $T_0 = O\left(k(\beta/\delta)^3\right)$ *such that,*

$$\forall t \geq T : \quad h_{t+1} \leq h_t \left(1 - \min\{\frac{\delta}{12\beta}, \frac{1}{2}\}\right)). \tag{42}$$

*Finally, under Assumption 1, we have that,*

$$\forall t \geq 1 : \quad \|\mathbf{V}_t - \mathbf{X}^*\|_F^2 = O\left(\frac{\beta^2}{\delta^3} h_t\right). \tag{43}$$

*Proof.* Result (41) follows from standard convergence results for the Frank-Wolfe method with line-search [12], and the fact that the Euclidean diameter of the Fantope $\mathcal{F}_{n,k}$ is $\sqrt{2k}$.

For the second part, observe that under Assumption 1, using the the $\beta$-smoothness of $f(\cdot)$, the quadratic growth result (Lemma 3) and (41), we have that for all $t \geq 1$,

$$\|\nabla f(\mathbf{X}_t) - \nabla f(\mathbf{X}^*)\|_F \leq \beta \|\mathbf{X}_t - \mathbf{X}^*\| \leq \beta \sqrt{\frac{2h_t}{\delta}} = O\left(\sqrt{\frac{k\beta^3}{t\delta}}\right).$$

Thus, for some $T_0 = O\left(k(\beta/\delta)^3\right)$ we have that,

$$\forall t \geq T_0 : \quad \|\nabla f(\mathbf{X}_t) - \nabla f(\mathbf{X}^*)\|_F \leq \frac{\delta}{3}.$$

Let us write the eigen-decomposition of $\nabla f(\mathbf{X}_t)$ as $\nabla f(\mathbf{X}_t) = \sum_{i=1}^{n} \lambda_i \mathbf{v}_i \mathbf{v}_i^{\top}$. Using Weyl's inequality for the eigenvalues we can write for every $t \geq T_0$,

$$\lambda_{n-k} - \lambda_{n-k+1} \geq \lambda_{n-k}(\nabla f(\mathbf{X}^*)) - \lambda_{n-k+1}(\nabla f(\mathbf{X}^*)) - 2\|\nabla f(\mathbf{X}) - \nabla f(\mathbf{X}^*)\|_F$$
$$\geq \delta - \frac{2\delta}{3} = \frac{\delta}{3}.$$

Thus, for all $t \geq T_0$, $\lambda_{n-k+1} < \lambda_{n-k}$ and the matrix $\mathbf{V}_t$ is uniquely defined and given by $\mathbf{V}_t = \sum_{i=n-k+1}^{n} \mathbf{v}_i \mathbf{v}_i^{\top}$. Using $\mathbf{X}_{t+1} = (1 - \eta_t)\mathbf{X}_t + \eta_t \mathbf{V}_t$, the smoothness of $f(\cdot)$, and the fact that $\eta_t$ is chosen via line-search, we have that,

$$\forall \eta \in [0, 1] : f(\mathbf{X}_{t+1}) \leq f(\mathbf{X}_t) + \eta \langle \mathbf{V}_t - \mathbf{X}_t, \nabla f(\mathbf{X}_t) \rangle + \frac{\eta^2 \beta}{2} \|\mathbf{V}_t - \mathbf{X}_t\|_F^2.$$

Subtracting $f(\mathbf{X}^*)$ from both sides and using Lemma 10 with gap $\delta_{\mathbf{X}} = \delta/3$, we have that for all $t \geq T_0$,

$$\forall \eta \in [0, 1] : \quad h_{t+1} \leq h_t + \frac{\eta}{2} \langle \mathbf{V}_t - \mathbf{X}_t, \nabla f(\mathbf{X}_t) \rangle + (\frac{\eta^2 \beta}{2} - \frac{\eta \delta}{12}) \|\mathbf{V}_t - \mathbf{X}_t\|_F^2$$
$$\leq (1 - \frac{\eta}{2})h_t + (\frac{\eta^2 \beta}{2} - \frac{\eta \delta}{12}) \|\mathbf{V}_t - \mathbf{X}_t\|_F^2,$$

where the last inequality follows from the convexity of $f(\cdot)$.

Now, if $\frac{\delta}{6\beta} \leq 1$, by setting $\eta = \frac{\delta}{6\beta}$ we have that $h_{t+1} \leq (1 - \frac{\delta}{12\beta})h_t$. Otherwise, $\delta > 6\beta$ and so, setting $\eta = 1$, we get that $h_{t+1} \leq \frac{1}{2}h_t$, which proves Result (42).

Finally, for the third part of the lemma, recalling that $\mathbf{V}_t$ and $\mathbf{X}^*$ are the projection matrices onto the span of the top $k$ eigenvectors of $-\nabla f(\mathbf{X}_t)$ and $-\nabla f(\mathbf{X}^*)$, respectively, using the well known Davis-Kahan $\sin\theta$ theorem (see for instance [27]), we have that for all $t \geq 1$,

$$\|\mathbf{V}_t - \mathbf{X}^*\|_F^2 \leq \frac{8\|\nabla f(\mathbf{X}_t) - \nabla f(\mathbf{X}^*)\|_F^2}{(\lambda_{n-k}(\nabla f(\mathbf{X}^*)) - \lambda_{n-k+1}(\nabla f(\mathbf{X}^*)))^2} \leq \frac{8\beta^2 \|\mathbf{X}_t - \mathbf{X}^*\|_F^2}{\delta^2} \leq \frac{16\beta^2 h_t}{\delta^3},$$

where the last inequality follows from the quadratic growth result, Lemma 3. Thus, Result (43) follows. $\square$