# OpenReview forum: "Local Linear Convergence of Gradient Methods for  Subspace Optimization via Strict Complementarity"
_NeurIPS.cc/2022/Conference — NeurIPS 2022 Accept_

### Official Review · Reviewer_FuJo · 2022-07-10

**Rating:** 6
**Confidence:** 5
**Soundness:** 3 good
**Presentation:** 4 excellent
**Contribution:** 3 good

**Summary:**

This paper studies the k-dimensional subspace problem, which is a nonconvex problem over the orthogonal matrix set. An efficient projection gradient method using QR decomposition is studied. The main contribution is that the strict complementarity is shown to be equivalent to the eigen-gap condition. Moreover, under the eigen-gap condition, the projection gradient method is shown to be linearly convergent.

**Questions:**

no

**Limitations:**

Not applied.

**Strengths And Weaknesses:**


Strengths:
This paper studies a general setting of k-dimensional subspace problem. The local linearly convergence under strict complementarity is interesting.

Weaknesses:
1. The symbol << is not standard, please use $\ll$ (\ll) instead
2. To use the KKT conditions, the slater's condition should be verified though it is obviously satisfied if k<n.
3. The numerical experiments do not show the linear convergence, since the linear convergence rates are  important results in this paper.
4. The linear convergence rate of projected gradient method for k-dimensional subspace problem is extensively studied in recent years. The strict complementarity or eigen-gap condition could be strong in some settings. For example, Liu et al. 2019 studies the quadratic problem on Stiefel manifold. But the do not need that the eigenvalues are distinct [Theorem 1, Liu et al. 2019]. Some discussions will be helpful.

Liu, H., So, A. M. C., & Wu, W. (2019). Quadratic optimization with orthogonality constraint: explicit Łojasiewicz exponent and linear convergence of retraction-based line-search and stochastic variance-reduced gradient methods. Mathematical Programming, 178(1), 215-262.

---

> ### Author Response · Authors · 2022-07-31
> **Reply**
>
> Dear reviewer,
>
> Thank you for your overall positive review.
>
> We now address the weakness you raised:
> 1+2: Thank you, we will correct these in final version.
> 3. The initial focus of the experiments was not on linear rates but showing the convergence of our methods from simple initializaitons. Nevertheless, while preparing the rebuttal we have verified that if we plot the Y-axis in log scale we clearly see linear convergence of our methods, and we will include these graphs in the final version.
> 4. Thank you very much for pointing us to this interesting work. We will add it to our discussion on related works. Please do note however that it concerns a very specific objective function (and one which seems nearly linear in the projection matrix XX^T), while here our goal is to address a much more general smooth convex setting. Moreover, they seem to establish (fast) convergence to critical points and not to global optimum. Generic linear convergence to global optimum is not likely even for their problem without further assumptions, since even for standard PCA, these are attainable only under a spectral gap assumption.
>
> Finally, please note that beyond the linear convergence rate, the main novelty in our work is the analysis of the gradient orthogonal iteration method which has both linear convergence, but also uses only a single QR operation per iteration, and as such is a novel non-linear extension of the classical QR iterations method for leading subspace computation (as in standard PCA). The analysis of this approach is the main novelty, and establishing its convergence by showing that it is a nearly lossless approximation of the exact-SVD-based projected gradient method. This comes with no little effort as might be indicated from Lemmas 4, 5, 9, among others. This is the key novelty. We believe that relating these two methods - a convex one and a nonconvex one is a very novel idea which could be interesting to further developments.

---

> > ### Author Response · Authors · 2022-08-08
> > **Reply?**
> >
> > Dear Reviewer FuJo,
> >
> > Have we answered your main concerns? If so, would you consider raising your score? Otherwise, we will be very happy to try and answer additional concerns.

---

### Official Review · Reviewer_tL19 · 2022-07-10

**Rating:** 5
**Confidence:** 4
**Soundness:** 3 good
**Presentation:** 3 good
**Contribution:** 2 fair

**Summary:**

For optimization problem of finding a $k$-dimensional subspace, iterative schemes derived from both non-convex and convex formulations are developed in the literature. In this paper, based on a strict complementarity condition, the authors proved local linear convergence of several schemes, including non-convex projected gradient descent and gradient orthogonal iteration. For the obtained convergence results, the starting point of the schemes need to be close enough to the solution.

**Questions:**

A small question, how to choose $k$ in practice?

The paper is not well-written overall, and the authors should polish the paper thoroughly.
 - line 5, what does "among others" mean?
 - line 7 "or ," to ", or"
 - line 12, the whole sentence "Our result ..." i guess needs rephrase.
 - line 24 "include among other..."
 - For the footnotes, please unify the starting letter in capital.
 - line 34"] however" to "]. However"
 - "aka" to "a.k.a."
 - line 59 "efficient implementation of will...", of what will?
 - line 81, "(2))" and ". $^3$"
 - line 188 $X \in \mathcal{F}\_{k}$, should be $\mathcal{F}\_{n,k}$?
 - page 8, Lemma 6,8,9 are not appear in the main paper.

**Limitations:**

Not applicable here.

**Strengths And Weaknesses:**

The main contribution or strength of the paper is theoretical analysis. That is, proving local linear convergence of non-convex projected gradient descent, gradient orthogonal iteration and Frank-Wolfe methods, which is based on connecting the non-convex model and convex one.

To me, there are several limitations of the obtained results:
 - Assumption 1 requires a parameter which cannot be verified before solving the problem. What happens if $k$ is chosen relatively large enough such that $\lambda_{n-k}=\lambda_{n-k+1} = 0$, should this be a problem to concern?
 - All the convergence rate theorems require the starting point of the numerical schemes to be closely enough to the solution, which is not discussed how to achieve this in practice. Hence making the local convergence result limited. It is also not elaborated what exactly is the "warm-start" strategy.
 - Lemma 3 the "quadratic growth" assumption basically is the weaker strong convexity assumption, the authors should discuss its connections/differences compared to existing approaches.

---

> ### Author Response · Authors · 2022-07-31
> **Reply**
>
> Dear reviewer,
>
> We first address the weaknesses you raised:
> 1. Assumption 1 requires a parameter that cannot be verified: First, note that the eigengap parameter is not an input to any of the algorithms. The parameter k is required to be set properly (so there is an eigengap) and here we have several answers:
> a. First we note that, as in classical PCA, the parameter k should be generally understood as part of the input to the problem (of course in practice it is an issue by itself how to set it). Classical numerical algorithms for PCA (such as classical QR iterations) are dependent on the choice of k, and its choice can affect dramatically their performance, and here we encounter a similar situation, so this should not be seen as a specific issue with our work.
> b. Indeed miss-specifying k is problematic for the gradient orthogonal iteration method since it requires strict complementarity to exactly hold. If there is no gap we cannot guarantee is convergence.
> c. This is not a critical problem for our PGD method since as Theorem 6 (see Appendix C) shows: PGD can handle gaps between lower-eigenvalues by increasing the rank parameter r, but guaranteeing only sublinear rates, not linear.  Moreover, as discussed in Remark 1 (line 180), for PGD we can verify easily whether it converges correctly or not (i.e., with a provable rate), by certifying that the low-rank projection is indeed the correct projection, which can helps us tune the parameters.
>
> 2.  Results are weak because warm-start is required: this is indeed true for the gradient orthogonal iteration and projected gradient, but this is very much expected, since these methods solve a nonconvex problem (they always maintain a rank-k matrix)! It should not be expected that, unless the problem is extremely well conditioned so the ball is so large, that they would converge rapidly from any initialization. This is also perfectly well aligned with previous works on generic nonconvex optimization (i.e., works that do not consider very specific models and data), see for instance [2]. Of course it may be expected that in practice these would work from very simple initializations (as we have in our experiments), but here we are interested in worst case performance.
> We refer you also to Figure 3 in the appendix that examine the empirical convergence from a random initialization and shows it to be considerably slower (sublinear).
> Moreover, in this work we wish to take a somewhat of a ''continuous optimization'' approach to subspace recovery (as opposed to statistically-motivated approaches), where we look for a quite general condition (strict comp.) that will render quite general problems (i.e., any smooth convex objective, as opposed to very specific objectives such as quadratic with data generated by some well known process), well-posed, and because of this we have these requirements. Do note that the Frank-Wolfe method converges globally, but with the price that it does not maintain a rank-k matrix, but a convex combination of such.
>
> 3. Quadratic growth is a well known property to allow for linear rates of first-order methods for convex problems and is indeed weaker than strong convexity, see for instance [14,19]. Since it is well-studied, it is beyond the scope of this paper to present it or discuss it in great detail, and we give the appropriate references.
>
> Questions:
> - How to choose k? As in standard PCA algorithms, k should be understood as part of the input. Our goal is to develop numerical methods given an appropriate k, much like the classical fast methods for computing the leading subspace in standard PCA. It is not part of the problem we set out to solve to find k.
>
> Typos: thank you for catching these!
>
>
> Given all of the above, we kindly ask you to seriously reconsider your score. We believe our novel approach of analyzing the gradient orthogonal iteration and the corresponding analysis, and its connection to strict complementarity, could be of interest to many working in these and related areas.

---

> > ### Comment · Reviewer_tL19 · 2022-08-07
> > **Reply to authors responses**
> >
> > Thanks for the responses.
> >
> > Regarding the choices of $k$, the authors said that "Our goal is to develop numerical methods given an appropriate k,", meanwhile for the strict complementarity, there is also a "$k$". I think these two $k$'s are not exactly the same, the "appropriate k" in general is larger than the later $k$? If this is true, and the "appropriate k" much larger (let's assume), is it possible to design some parameter continuation strategy to gradually reduce the value of the "appropriate k", based on the complementarity condition?

---

> > > ### Author Response · Authors · 2022-08-07
> > > **reply**
> > >
> > > It is the same k. If our goal is to find a projection matrix onto a k-dimensional subspace that minimizes a convex and smooth loss f(), then strict complementarity is defined in terms of the eigengap between the k-largest and (k+1)-largest eigenvalues. With this respect our work does not deal with how to choose k. This is very similar to classical PCA where we want to extract the leading k-dimensional subspace of the covariance matrix: it is well conditioned if for the chosen k there is indeed a substantial gap in the covariance matrix  between the k and (k+1) largest eigenvalues.
> > >
> > > A one exception is our Theorem 6, mentioned also in our response, that deals with the case in which strict complementarity does not hold, or holds with a negligible gap. In this case in may be the case that the solution to the convex relaxation is no longer a rank-k matrix as desired. However, this theorem shows that by considering higher-rank matrices, with rank=r>k, it might still be possible to run PGD in an efficient manner, i.e., using only low-rank SVD to compute the projection (rank-r SVD instead of a full rank SVD) and the optimal solution may still be low rank, even if with rank larger than k.
> > >
> > > We hope this helps. We are very happy to help clarify this issue further.

---

> > > > ### Comment · Reviewer_tL19 · 2022-08-08
> > > > **Reply**
> > > >
> > > > Many thanks for the replies, and I have raised my score.

---

### Official Review · Reviewer_Lsrb · 2022-07-11

**Rating:** 6
**Confidence:** 4
**Soundness:** 3 good
**Presentation:** 3 good
**Contribution:** 2 fair

**Summary:**

In this work, the authors studied the problem of finding a $k$-dimensional subspace problem: $\min f(\mathbf{Q}\mathbf{Q}^T)\ \text{s.t.}\ \mathbf{Q}^T\mathbf{Q} = \mathbf{I}$, where $f$ is convex and smooth. They studied the relationship between the convex relaxation-based method and the non-convex method with the gradient orthogonal iterations under a strict complementarity assumption. Based on this, they showed that the non-convex gradient method converges locally with a linear rate. They also showed the linear convergence of the non-convex projected gradient method and the Frank-Wolfe method for solving this problem.

**Questions:**


(${\bf 1}$)  The authors mentioned that strictly complementarity is closely related to the error-bound condition (see Ref [29, 6]). It is known that the error-bound condition also implies linear convergence of the first-order method under some mild conditions. Does Assumption 1 imply the error bound of Problem (4)? It would be great if the authors elaborate on the relationship between them.

(${\bf 2}$) According to Theorems 3 and 4, the authors only showed the linear convergence of function values. According to the results in [29] and $\textbf{Attouch and Bolte (2009)}$, they can show linear convergence of the sequence generated by the studied methods. Could the authors show the linear convergence of the sequence?

$\textbf{Attouch and Bolte (2009)}$: On the convergence of the proximal algorithm for nonsmooth functions involving analytic features. Mathematical Programming, 116(1), 5-16.

(${\bf 3}$) Lemma 4 shows quadratic growth of Problem (4). According to [6], quadratic growth is equivalent to the error bound under a mild condition. Maybe, the following proofs could be simplified using the results in [6] and [29] when the error bound of Problem (4) is available. Please check it.

**Limitations:**

Yes

**Strengths And Weaknesses:**

Strengths: (${\bf 1}$) Subspace recovery is a fundamental problem in machine learning. This paper studied the convergence rate of different methods for solving  the subspace problem: $\min f(\mathbf{Q}\mathbf{Q}^T)\ \text{s.t.}\ \mathbf{Q}^T\mathbf{Q} = \mathbf{I}$. Unlike existing approaches that assume $f$ admits a special structure or considers an underlying generative model, they proposed a deterministic condition, i.e., strict complementarity, which seems new and interesting.

(${\bf 2}$)  Overall, the paper is well-written and easy to follow. I think that the authors made good enough technical contributions to the convergence analysis of the first-order methods for solving subspace optimization problems.

(${\bf 3}$)  This work bridges convex and non-convex methods for subspace optimization problems via a strict complementarity condition. In particular, linear convergence can be established through this bridge.

Weaknesses: (${\bf 1}$)  In Assumption 1, the authors imposed an eigen-gap condition, i.e., $\lambda_{n-k}(\nabla f(\mathbf{X}^*)) - \lambda_{n-k+1}(\nabla f(\mathbf{X}^*)) \ge \delta$ for some $\delta > 0$, on an optimal solution $ \mathbf{X}^*$ to the problem: $\min f(\mathbf{X})\ \text{s.t.}\  \mathbf{I} \succeq \mathbf{X} \succeq \mathbf{0},\ \mathrm{Tr}( \mathbf{X}) = k$. From the authors' introduction, this condition is closely related to the complementarity condition of the KKT system of Problem (4) (see Theorem 2). It looks strange that studying convergence rate of a method for solving a problem by imposing condition onto its optimal solution. On the other side, the authors mentioned the error-bound condition. According to [29], the error-bound condition holds for many general problems.  However, the authors in this work cannot give a concrete example so that Assumption 1 holds like [29].

(${\bf 2}$)  In general, the numerical simulations are not convincing. For example, in the left figure of Figure 1 in Section 4, the y-axis should be plotted on a log scale. The current scale cannot demonstrate the linear convergence of the tested method. Besides, the authors should also report the convergence rate of the Frank-Wolfe method to support Theorem 5.

---

> ### Author Response · Authors · 2022-07-31
> **Reply**
>
> Dear reviewer,
>
> Thank you for your overall positive review.
>
> We now address the weakness you raised:
> 1. We think you might have misunderstood [29]: the condition [29] has for which their error bound holds for matrix problems (they consider nuclear norm regularization) is very similar to the one we have and is in fact equivalent to strict complementarity for their nuclear norm-regularized model.
> As you write yourself, our purpose is to take a ``continuous optimization''  approach which seeks a condition that will render. quite general problems well conditioned for first-order methods. This is also the reason we bring the numerical evidence to show that for two classical robust PCA models this condition seems to hold very well. In works [5,9] is was shown that if such a condition does not hold then the convex relaxation is ill-posed since it is brittle under arbitrary small perturbations. Such results could be extended to our subspace recovery problem as well, and we shall comment on it in the final version.
>
> 2. Indeed when we plot the Y-axis in log-scale (which we have done during the preparation of this rebuttal) we clearly see a linear convergence rate, we shall add this to the final version.
> Please understand, that as you yourself write ''I think that the authors made good enough technical contributions to the convergence analysis of the first-order methods for solving subspace optimization problems'', we believe our theoretical contribution is strong enough and doing lots of experiments is beyond our interest which is mainly in theoretical analysis.
> The main purpose of the graphs is not to compare the methods, but mainly to show that indeed simple initializations start PGD already in the regime in which it produces only rank-k iterates, and that the gradient orthogonal iteration indeed converges very similarly to PGD. We will also strongly consider adding the plots of Frank-Wolfe, this should not be a problem, just not our main interest.
>
> Answers to questions:
> 1. We believe that it does. The assumption that [29] has for the nuclear norm-regularized matrix problem is exactly equivalent to strict complementarity for their problem and they use it to obtain their error-bound.
>
> 2. Yes definitely. Using the quadratic growth property (Lemma 3), we can relate the convergence in function value to the convergence of the sequence to the optimal solution.
>
> 3. The bulk of our analysis is not in obtaining the linear rates, such analysis is standard and simple. Our main novelty and technical effort is in proving that the gradient orthogonal iteration, which only performs a single QR operation per iteration, indeed converges correctly, by establishing that it approximates sufficiently well the steps of PGD -- for which we prove that near the minimizer it produces only low-rank matrices. That is the bulk of the analysis, not the linear rate.

---

> > ### Author Response · Authors · 2022-08-08
> > **Reply?**
> >
> > Dear Reviewer Lsrb,
> >
> > Have we answered your main concerns? If so, would you consider raising your score? Otherwise, we will be very happy to try and answer additional concerns.

---

### Official Review · Reviewer_mnGb · 2022-07-12

**Rating:** 4
**Confidence:** 4
**Soundness:** 3 good
**Presentation:** 3 good
**Contribution:** 2 fair

**Summary:**

The paper considers the problem of minimizing a convex function f of symmetric n-by-n matrices X over the space P(n,k) of orthogonal projectors on all k-dimensional subspaces of R^n. A special case is PCA but the formulation allows for generalized forms of PCA (robust, sparse, etc). The natural iterative method to approach this problem is projected gradient descent applied to the original problem min f(X) s.t. X\in P(n,k), in which we need to project on P(n,k) (which requires SVD) in every iteration. The paper proposes a low-rank iterative method applied to the problem min f(Q*Q') s.t. Q'*Q=I, Q is n-by-k, which requires only one QR factorization in every iteration (this can be seen as approximating SVD by doing only one iteration of the orthogonal iterations method). To analyze this method, the paper also considers an SDP relaxation of the original problem, in which P(n,k) is replaced with its convex hull F(n,k) (the Fantope), which has an SDP description.

The contribution of the paper is a theoretical analysis of the low-rank iterative method. The key concept is  the "eigengap" of a solution X to the SDP relaxation, which is the difference between the k-th and (k+1)-th largest eigenvalues of the gradient \nabla f(X) (a symmetric matrix). It is proved that existence of an optimal solution to the SDP with non-zero eigengap is equivalent to strict complementarity of a pair of primal-dual solution to the SDP (Theorem 1) and implies that the SDP has a unique optimum with rank k (hence, the SDP relaxation is tight) (Theorem 2).

The main contribution is a convergence result for the low-rank iterative method (Theorem 3), which says the following: If the SDP relaxation have a non-zero eigengap and the initial point Q1 of the method is such that Q1*Q1' is within a given (small) ball around the optimal SDP solution, then the method converges linearly to the global optimum of the original problem.

Almost all the rest of the paper is devoted to a sketch of the proof of Theorem 3. At the end of the paper, there are brief numerical experiments on random data, showing that the convergence rates of the projected-gradient method and the low-rank method are almost the same (but, as written above, the latter needs only one QR factorization rather than SVD factorization in each iteration).

**Questions:**

1. Can it happen (assuming a nonzero eigengap assumption) that the rank of Z_{t+1} is smaller than k in (3)? If so, the QR factorization must do some random choice. Is the iteration valid then?

2. Can fixed points of iterations (3) be described in some simple way, such as a closed formula?

3. Does there hold also the opposite implication in Theorem 1, i.e., is it true that if a (optimal) solution to the SDP (4) has rank k then it has a nonzero eigengap?

4. Does it hold that if the SDP relaxation (4) is tight (i.e., some of its optimal solutions has rank k), then problem (1) has no local minima that are not global minima (where "local minimum" has the obvious meaning here as a local minimum of a function on a set)? Note, this could be easily supported or disproved by numerical simulations.

5. A very related question: Suppose some optimal solution to the SDP has a nonzero eigengap but the initial point Q1 to method (3) does not satisfy the other assumption of Theorem 3 (i.e., Q1*Q1' is not within the given ball around the optimal solution to the SDP). Does it mean that method (3) may not converge at all, can converge to the global optimum but with sub-linear rate, or converge to a point that is not the global optimum? Some of these options could (and should) also be supported or disproved by numerical experiments.

**Limitations:**

The impact of the main theoretical result (Theorem 3) is rather weak, see above.

**Strengths And Weaknesses:**

This is an interesting formulation of the low-rank method, which requires only one QR (rather than SVD) factorization per iteration.

The theoretical analysis is interesting. However, its novelty is only incremental, given the prior work [5] and [8,9].
Moreover, its impact is rather weak: if the initial point of the low-rank method is not within a small ball of the optimal solution (which we of course do not know without solving the SDP), the results (Thm 3) give no guarantees of convergence to the global optimum.

The text is easy to understand, but its clarity and organization should be improved. In particular, it would be more helpful if all longer proofs were moved to a supplement and the saved space was used to explain the consequences of the theoretical results in more detail. E.g., the questions I raise in the "Questions" part could be discussed.

The numerical experiments are very brief. I believe that more numerical experiments might clarify some issues as I suggest in the "Questions" part. Moreover, experiments on data from real applications might better show the properties of the method.

In summary, I think that the main ideas are interesting but I am not sure if the current contribution and its form are good enough for this conference. I believe that giving the paper more time and effort to develop would result in a better and more mature paper.

Minor remarks:
- Perhaps, it would be useful in the theoretical analysis to consider one more iterative method, namely projected gradient method applied to the problem min f(Q*Q') s.t. Q'*Q=I, Q is n-by-k. This method requires orthogonal projection to the space of n-by-k matrices with orthonormal columns (i.e., the nearest linear isometry problem), which requires SVD. Perhaps, method (3) can be more directly seen as an approximation to this method than to method (2). But I may be wrong, this is just a suggestion.
- I'd use a different symbol than \partial for the total derivative in (3), such as {\rm d}.
- Typo in Lagrange function below line 101, (X) should be f(X).
- In Definition 1, I believe that disjunction (\vee) should be conjunction (\wedge).

---

> ### Author Response · Authors · 2022-07-31
> **Reply**
>
> Dear reviewer,
>
> We first address the weaknesses you've raised:
> 1. Novelty is incremental w.r.t. [5] and [8,9]: We believe there might be a miss-understanding here. Indeed the quadratic growth result which yields the linear convergence rate is not very novel and we did not try to pretend that it is. The novelty is in the fact that all these previous results rely on SVD computation. Each such computation, when implemented efficiently, requires to iteratively perform multiple QR iterations. Here on the otherhand we have method with same convergence rate using a single QR operation per iteration. The analysis of this approach is the main novelty, and establishing its convergence by showing that it is a nearly lossless approximation of the exact-SVD-based projected gradient method. This comes with no little effort as might be indicated from Lemmas 4, 5, 9, among others. This is the key novelty. We believe that relating these two methods - a convex one and a nonconvex one is a very novel idea which could be interesting to further developments.
> Additional novelty is in the proof of Theorem 6 (in Appendix C) that gives sublinear rates for projected gradient in case exact strict complementarity does not hold, but some relaxed notions of it. This is a significant and non-trivial at all extension of [9].
>
> 2. Results are weak because warm-start is required: this is indeed true for the gradient orthogonal iteration and projected gradient, but this is very much expected, since these method solve a nonconvex problem (they always maintain a rank-k matrix)! It should not be expected that, unless the problem is extremely well conditioned so the ball is so large, that they would converge rapidly from any initialization. This is also perfectly well aligned with previous works on generic nonconvex optimization (i.e., works that do not consider very specific models and data), see for instance [2]. Of course it may be expected that in practice these would work from very simple initializations (as we have in our experiments), but here we are interested in worst case performance.
> We refer you also to Figure 3 in the appendix that examine the empirical convergence from a random initialization and shows it to be considerably slower (sublinear).
> Moreover, in this work we wish to take a somewhat of a ''continuous optimization'' approach to subspace recovery (as opposed to statistically-motivated approaches), where we look for a quite general condition (strict comp.) that will render quite general problems (i.e., any smooth convex objective, as opposed to very specific objectives such as quadratic with data generated by some well known process), well-posed, and because of this we have these requirements.
> Do note that the Frank-Wolfe method converges globally, but with the price that it does not maintain a rank-k matrix, but a convex combination of such.
>
> 3. Experiments: We are mainly interested in the theory of efficient first-order optimization methods. PGD is often the method of choice for smooth convex objectives and so we focus on demonstrating on two robust pca models that indeed using very simple initalizations that indeed PGD converges correctly while maintaining only a rank-k matrix, and that the gradient orthogonal iteration method indeed approximates the steps of PGD very well. If we present the Y-axis in log-scale (as we shall do in our final version) then it also becomes very clear that indeed both methods converge linearly. Doing more comprehensive experiments is beyond the scope of our work since we are mostly interested in understanding the theory and per the discussion above, we think we have sufficient theoretical results.
>
> Answers to questions:
> 1. Not it cannot happen that the rank is smaller than k, we shall comment on it in the paper.
> 2. There is no simple way to write this update as a closed formula.
> 3. This direction is not true. It can be the case that there exists an optimal solution of rank k and eigen-gap is zero, for instance if the gradient is zero at the optimal solution.
> 4. We do not think its true in general: consider the simple problem of computing the leading eigenvector of a symmetric matrix, which corresponds to a linear function f in our setting. Any eigenvector is a stationary point of the nonconvex fomulation, while the convex relaxation is always tight.
> 5. We refer you to Figure 3 in the appendix which shows such an experiment. It can be seen that while we have convergence from a random initialization, it is much slower, hinting that in practice indeed sublinear convergence may be expected, but this well require very different arguments to prove and is beyond the scope of this work.
>
> Given all of the above, we kindly ask you to seriously reconsider your score. We believe our novel approach of analyzing the gradient orthogonal iteration and the corresponding analysis, and its connection to strict complementarity, could be of interest to many.

---

> > ### Author Response · Authors · 2022-08-08
> > **Reply?**
> >
> > Dear Reviewer mnGb,
> >
> > We would be very happy to know if the above response answers your main concerns and will be also very happy do further discuss remaining concerns.

---

### Author Response · Authors · 2022-08-03
**Revised version**

Dear Reviewers and AC,

Quite embarrassingly, we were not aware until just now that there is an option to revise our submission during the rebuttal period and so we did not plan for it (or allocate time for it).

Nevertheless, we have uploaded a slightly revised version in which we address an issue raised by Reviewer Lsrb and Reviewer FuJo, regarding empirical evidence in support of the linear convergence rates. In the revised version we added plots (left panels in Figure 1 and Figure 2) which plot the convergence rate w.r.t. function value in log-scale, and clearly demonstrate the linear convergence of projected gradient and gradient orthogonal iteration methods.

We shall address the additional issues in the final version.

If there are additional question, we will be happy to answer.

---

### Meta-Review · Area_Chair_AqXy · 2022-08-24

**Recommendation:** Accept
**Confidence:** Certain

**Metareview:**

The submitted work presents a local linear convergence guarantee for a projected gradient descent (PGD) algorithm on an explicit parameterization of the Stiefel manifold. Such a guarantee is easy to make if the convex objective f is assumed to be strongly convex. Instead, this work considers allowing f to be non-strongly convex. Under a strict complementarity assumption, which this paper shows is equivalent to an eigen-gap condition, the authors prove that the problem enjoys a standard quadratic growth condition that allows PGD to converge at a linear rate.

Reviewers Lsrb, tL19, Fujo concur that the theoretical contribution is worthy of publication. The past few years have seen a large number of local linear convergence guarantees by directly optimizing the factor matrix $U$ in the low-rank factorization $X=UU^T$, but all of these work have assumed some notion of strong convexity or restricted strong convexity. Indeed, I remark here that local linear convergence is actually lost in many of these cases (e.g. matrix sensing) if the objective f is not (restrictedly) strongly convex. In comparison, the present work allows f to be an arbitrary smooth convex function, while showing that local linear convergence is surprisingly still possible under a strict complementarity condition.

However, the impact of the work is obfuscated by repeated assertions to the practical aspects of the proposed algorithm, which in my opinion are difficult to defend. The authors repeatedly assert that their nonconvex algorithm requires only a single QR decomposition, and therefore "much faster and simpler to implement". This may be the case, but the actual reduction in the number of QR decompositions is only a logarithmic factor $O(\log(1/\epsilon))$ under the eigen-gap assumption. On the other hand, global convergence is lost with the nonconvex formulation, and random initialization leads to sublinear convergence in practice. Reviewer mnGb remarks that the numerical experiments are very brief, and do not make a strong case for the practical aspects of the algorithm.

Nevertheless, the technical novelty of the analysis pushes this paper towards acceptance. In the camera-ready version, the authors are advised to:
* Revise their summary of contributions to better compare with existing techniques in the literature, as outlined by Reviewers Lsrb, tL19, Fujo;
* Expand on their experimental section to answer the questions posed by Reviewer mnGb on global convergence, the existence of bad local minima. Answers to these questions can and should be supported or disproved by numerical experiments.



**Award:**

No

---

### Decision · Program_Chairs · 2022-09-14

Accept